# DECISION MAKING UNDER IMPERFECT RECALL: ALGORITHMS AND BENCHMARKS

## ABSTRACT

In game theory, imperfect-recall decision problems model situations in which an agent forgets information it held before. They encompass games such as the "absentminded driver" and team games with limited communication. In this paper, we introduce the first benchmark suite for imperfect-recall decision problems. Our benchmarks capture a variety of problem types, including ones concerning privacy in AI systems that elicit sensitive information, and AI safety via testing of agents in simulation. Across 61 problem instances generated using this suite, we evaluate the performance of different algorithms for finding first-order optimal strategies in such problems. In particular, we introduce the family of regret matching (RM) algorithms for nonlinear constrained optimization. This class of parameter-free algorithms has enjoyed tremendous success in solving large two-player zero-sum games, but, surprisingly, they were hitherto relatively unexplored beyond that setting. Our key finding is that RM algorithms consistently outperform commonly employed first-order optimizers such as projected gradient descent, often by orders of magnitude. This establishes, for the first time, the RM family as a formidable approach to large-scale constrained optimization problems.

## 1 INTRODUCTION

*Imperfect-recall* decision problems capture settings in which an agent can forget previously acquired information (Rubinstein, 1998). Humans are prone to forgetting, but why should we design or model AI agents with imperfect recall? Several applications have already garnered considerable attention. A prominent one concerns *team games*—strategic interactions in which multiple players strive toward a common objective. A central challenge there stems from the fact that communication or coordination between players is often infeasible or expensive (Von Stengel & Koller, 1997; Zhang et al., 2022; 2023; Basilico et al., 2017). The inherent asymmetry of information between the players can then be captured as a single meta-player that faces an imperfect-recall decision problem. Another influential application revolves around real-world problems that are too large to handle, and therefore need to be compressed in a *game abstraction*. Abstractions with imperfect recall, in particular, form a key component of state-of-the-art algorithms for game solving (Waugh et al., 2009; Kroer & Sandholm, 2014; 2016; Waugh, 2009; Lanctot et al., 2012; Benjamin & Lanctot, 2024).

With the rapid proliferation of AI, questions of trustworthiness have also been brought to the fore. Institutions and governing bodies test and evaluate AI agents extensively in simulated environments to verify their performance and safety upon deployment (Pan et al., 2023; Kinniment et al., 2024). This hinges on the assumption that the agent cannot distinguish between whether it is acting in the real world or in a simulated environment; otherwise, it may obscure its intentions temporarily during testing to secure deployment in the real world (Kovařík et al., 2025a). This has happened, for example, in the infamous Volkswagen (multi-billion-dollar) emission scandal in 2015, which centered on the surreptitious use of software in some Volkswagen diesel vehicles to detect emission testing. Consequently, effective evaluation protocols hinge on the agent not being able to make such distinctions, which also requires that it *forgets* whether it has acted in a simulated environment before or not. Kovařík et al. (2023) introduced the framework of *simulation games* to address such problems (*cf.* Chen et al., 2024; Oesterheld, 2019; Cooper et al., 2025).

Last but not least, imperfect recall is critical in the ubiquitous cases where an AI system handles private information. Data privacy laws are predicated on selectively relinquishing sensitive informa-

tion, a premise exemplified by the European Parliament and Council of the EU (2016) GDPR "right to be forgotten" act. As an example, consider a medical AI system tasked with identifying suitable candidates for blood donation. Potential candidates would be reluctant to share confidential information about their health status—HIV status, medical history, etc.—*unless* the AI has been designed to delete any knowledge regarding patients that were deemed unsuitable, thus exhibiting imperfect recall. In another example coming from the economics of innovation, Arrow's disclosure paradox (Arrow, 1962) describes the perennial challenge in which an inventor must reveal information about a new idea to secure funding, but such disclosure risks expropriation (Nelson, 1959). Stephenson et al. (2025) propose and investigate delegating decision making to an imperfect-recall AI agent as one possible solution to this dilemma. Taken together, it stands to reason that decision problems with imperfect recall will play a key role in AI going forward.

**Our Contributions**

Decision making under imperfect recall, and specifically *absentmindedness*, have been extensively studied since the early years of game theory (cf. Kuhn, 1953, and other work discussed in the appendix). So far, this has been done with pen and paper. Our work is the first to develop an empirical framework for decision making under imperfect recall through a flexible suite of benchmarks. Specifically, we construct three key types of parametrized tabular problems motivated by the prevalent applications discussed above. We refer to them as simulation problems (Section 4.1), subgroup detection problems under privacy constraints (Section 4.2), and random problems (Section 4.3).

In the second part of the paper, we turn to designing algorithms for solving such problems at scale, and evaluating them on 61 generated problem instances from our benchmark suite. First, we need to specify what constitutes a solution. The most natural objective is to identify an (*ex ante*) optimal strategy. Unfortunately, this is tantamount to finding a global optimum of a polynomial optimization problem, which is NP-hard (Koller & Megiddo, 1992). This is not just a theoretical obstacle: in our experiments, we find that a popular commercial solver for nonlinear optimization—namely, Gurobi—fails to converge beyond tiny instances. Thus, it is essential to relax our solution concept to tackle large problems. Following a recent line of work, we focus on computing *Causal Decision Theory* (CDT) equilibria (Lambert et al., 2019; Tewolde et al., 2023), which can be viewed as the set of KKT points—equivalently, first-order optima—of the underlying optimization problem. As such, CDT equilibria are amenable to scalable first-order optimizers such as projected gradient descent (PGD), which we use as the main baseline. As expected, our experiments show that PGD scales to much larger problem instances than Gurobi.

More surprisingly, our key algorithmic finding is that PGD and its variants are far from the best approach for this class of problems. In particular, we introduce the family of *regret matching (RM)* algorithms for imperfect-recall decision problems—which is tantamount to nonlinear constrained optimization. This class of algorithms has already enjoyed tremendous success in the restricted setting of solving large (two-player) zero-sum games, being at the heart of many milestone results (Moravčík et al., 2017; Brown & Sandholm, 2018; 2019). RM goes back to the pioneering work of Blackwell (1956) that laid the foundations of online learning. Part of its appeal lies in the fact that it is parameter-free. Yet, it has remained unexplored beyond zero-sum games, modulo some exceptions which are discussed in the the related work section that can be found in the appendix. For example, the imperfect-recall games that can arise in the game abstraction literature are, by design, very structured, and intend to accurately model an underlying perfect-recall game. Our work, on the other hand, focuses on solving decision problems that *inherently* feature imperfect recall, and tackles the problem in its full generality.

We pursue this direction and find that the RM family of algorithms consistently outperform PGD and its variants in terms of speed of convergence, typically by many orders of magnitude. This establishes for the first time that RM-based algorithms are formidable first-order optimizers. Further, not only are RM algorithms faster to converge, but they also consistently attain values at least as large as PGD, and oftentimes strictly larger. Both of those findings are surprising. The fact that RM and its variants perform remarkably well in two-player zero-sum games is a poor indicator of what would happen in constrained nonlinear optimization since the latter problem class is fundamentally harder.

We will make our benchmarks and code publicly available. Taken as a whole, we lay the groundwork for automatically analyzing decision problems under imperfect recall, beyond the toy instances that have been analyzed in the past (Kovařík et al., 2023; 2025b; Chen et al., 2024; Berker et al., 2025).

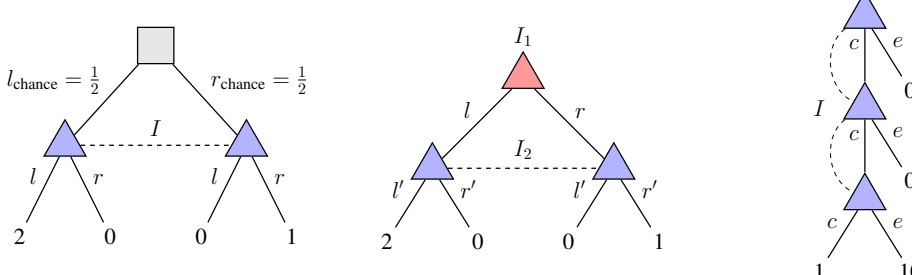

Figure 1: Three tree-form decision problems, discussed in the paragraph "Infosets and imperfect recall". The latter two are of imperfect recall. The rightmost further exhibits absentmindedness.

## 2 PRELIMINARIES

We begin by introducing imperfect-recall sequential decision making (Section 2.1). In Section 2.2, we then describe some standard solution concepts and known results concerning their computation.

### 2.1 DECISION PROBLEMS UNDER IMPERFECT RECALL

We operate under the standard framework of *tree-form* (aka. extensive-form) decision problems; for additional background, we refer to Piccione & Rubinstein (1997) and Fudenberg & Tirole (1991).

**Definition 1.** *A* tree-form decision problem*, denoted by $\Gamma$, consists of*

1. *A rooted tree with node set $\mathcal{H}$ and edges labeled with* actions. *The decision process starts at the root node $h_0$ and ends at some leaf node, also called* terminal node. *We denote the terminal nodes in $\mathcal{H}$ as $\mathcal{Z}$ and the set of actions available at a nonterminal node $h \in \mathcal{H} \setminus \mathcal{Z}$ as $A_h$.*
2. *An assignment partition $\mathcal{H} \setminus \mathcal{Z} = \mathcal{H}^* \sqcup \mathcal{H}^{(c)}$ of nonterminal nodes to either (i) the player of the decision problem or (ii) the* chance *"player" $c$ that models exogenous stochasticity. At each chance node $h \in \mathcal{H}^{(c)}$, actions are sampled according to a fixed distribution $\mathbb{P}^{(c)}(\cdot \mid h)$ over $A_h$.*
3. *A utility function $u : \mathcal{Z} \to \mathbb{R}$ that specifies the payoff the player receives when the decision process finishes at a terminal node.*
4. *A collection $\mathcal{I}$ of information sets (*infosets*) that partitions the player's decision nodes as $\mathcal{H}^* = \sqcup_{I \in \mathcal{I}} I$. We require $A_h = A_{h'}$ for all nodes $h, h'$ of the same infoset $I$. Therefore, the infoset $I$ has a well-defined action set $A_I$.*

**Infosets and imperfect recall**  The infoset structure captures the presence of imperfect information. Nodes of the same infoset are indistinguishable for the player. One possible source of imperfect information is the fact that the player is sometimes unable to observe the actions of another player, as illustrated in Figure 1 (left) w.r.t. the chance player. The player may also forget information it previously acquired; in that case, we say that the player exhibits *imperfect recall*. In Figure 1 (middle), for example, it cannot recall whether it played the left ($l$) or right ($r$) action in the past. A particular manifestation of imperfect recall is *absentmindedness*: the player in Figure 1 (right) cannot discern at a decision node whether it has been in the same situation (*i.e.* infoset) before.

Formally, each node $h \in \mathcal{H}$ in the decision tree is uniquely associated with a history path $\text{hist}(h)$, comprising a sequence of alternating nodes and actions from the root $h_0$ to $h$. On the path $\text{hist}(h)$, the player only encounters the sequence $\text{seq}(h)$ comprising infosets visited and actions taken by the player itself. We say an infoset $I$ is of *perfect recall* if for all nodes $h, h' \in I$, we have $\text{seq}(h) = \text{seq}(h')$—informally, the player can reconstruct the sequence $\text{seq}(h)$ from observing $I$ alone. Otherwise, it exhibits imperfect recall. The infoset $I$ exhibits *absentmindedness* if there exist distinct nodes $h, h' \in I$ with $h \in \text{hist}(h')$. By extension, we say that the entire decision problem has imperfect recall (resp. absentmindedness) if it is the case for at least one of its infosets.

**Strategies**  A *(behavioral) strategy $x$* for the player in $\Gamma$ specifies for any infoset $I$ of $\Gamma$ a probability distribution over the available actions at $I$. Upon reaching $I$, it will draw an action randomly according to that probability distribution, henceforth called *randomized action* and represented as

$\boldsymbol{x}(\cdot \mid I)$. Denoting the probability simplex at $I$ by $\Delta(A_I)$, a strategy $\boldsymbol{x}$ is an element of the product of simplices $\mathcal{X} := \bigtimes_{I \in \mathcal{I}} \Delta(A_I)$. A *pure strategy* is a tuple in $\bigtimes_{I \in \mathcal{I}} A_I \subset \mathcal{X}$.

**Reach probabilities and utilities**    The *reach probability* $\mathbb{P}(\bar{h} \mid \boldsymbol{x}, h)$ is the probability of arriving at node $\bar{h} \in \mathcal{H}$ when the player plays according to the strategy $\boldsymbol{x}$ and is currently at node $h \in \mathcal{H}$. This is the product of probabilities of the actions on the path from $h$ to $\bar{h}$ when $h \in \text{hist}(\bar{h})$, and $0$ otherwise. The expected utility of the player from being at node $h \in \mathcal{H} \setminus \mathcal{Z}$ and following profile $\boldsymbol{x}$ is $U(\boldsymbol{x} \mid h) := \sum_{z \in \mathcal{Z}} u(z) \cdot \mathbb{P}(z \mid \boldsymbol{x}, h)$. We will simplify our notation for the special case where the player is at the root node $h_0$ by defining $\mathbb{P}(h \mid \boldsymbol{x}) := \mathbb{P}(h \mid \boldsymbol{x}, h_0)$; similarly, we define the function $U : \mathcal{X} \to \mathbb{R}$ as $U(\boldsymbol{x}) := U(\boldsymbol{x} \mid h_0)$, mapping a profile $\boldsymbol{x}$ to its expected utility with respect to the root node. For example, the utility function in Figure 1 (right) reads $U(\boldsymbol{x}) = 1 \cdot \boldsymbol{x}(c \mid I)^3 + 10 \cdot \boldsymbol{x}(c \mid I)^2 \cdot \boldsymbol{x}(e \mid I)$. More generally, there is a strong connection between decision making under imperfect recall and polynomial maximization over a product of simplices.

**Theorem 1** (Equivalence to Polynomial Optimization; Gimbert et al., 2020; Tewolde et al., 2023)**.**

1. *Maximizing utility in a decision making problem under imperfect recall is captured by maximizing $U$—a polynomial function in $\boldsymbol{x}$—over the product of simplices $\mathcal{X}$.*
2. *Any constrained maximization problem $\max_{\boldsymbol{x} \in \mathcal{X}} p(\boldsymbol{x})$ of a polynomial function $p$ over a hypercube or a product of simplices $\mathcal{X}$ can be captured by a utility maximization problem of a decision making instance* à la *Definition 1.*

We expand on Theorem 1 in the appendix, and visualize the latter construction concisely on the optimization example $\max 2x^2 y - 3xyz$ s.t. $0 \leq x, y, z \leq 1$. In Theorem 1, the presence of absentmindedness in an infoset $I$ corresponds to a higher-degree dependency of $U$ in $\boldsymbol{x}(\cdot \mid I)$. This can necessitate the use of randomized actions in optimal strategies; see Figure 1 (right). All in all, Theorem 1 shows that the goal of this paper to study how to solve a decision making problem under imperfect recall is—at least in theory—equivalent to polynomial optimization over simplices.

## 2.2 Solution concepts

We call a strategy $\boldsymbol{x}^*$ $\epsilon$-optimal (for $\epsilon \geq 0$) if $U(\boldsymbol{x}^*) \geq U(\boldsymbol{x}) - \epsilon$ for all $\boldsymbol{x} \in \mathcal{X}$. Unfortunately, it is computationally hard to find an $\epsilon$-optimal strategy; or much simpler, to decide (up to a *constant* precision $\epsilon$) whether a particular value $v \in \mathbb{R}$ can be reached.

**Proposition 2** (Koller & Megiddo, 1992; Tewolde et al., 2023)**.** *Let $0 < \epsilon < 1/8$. Given a decision problem $\Gamma$ and a target value $v \in \mathbb{R}$, it is NP-complete to distinguish between whether $\Gamma$ admits a strategy $\boldsymbol{x} \in \mathcal{X}$ with $U(\boldsymbol{x}) \geq v$ or whether all strategies $\boldsymbol{x} \in \mathcal{X}$ satisfy $U(\boldsymbol{x}) \leq v - \epsilon$.*

In light of these theoretical limitations—which will be supported by our empirical findings—past work has studied relaxed solution concepts. One such notion, the *causal decision theory* (*CDT*) equilibrium, is particularly amendable to optimization algorithms. The basic idea behind the CDT equilibrium is that whenever the player must take an action at an information set, it considers whether it is beneficial for it to deviate *just this one time* from what $\boldsymbol{x}$ prescribes. To determine the expected gain from such a deviation, it assumes that it will continue to play according to $\boldsymbol{x}$ at all other decision nodes of the decision problem. (We provide further background on CDT equilibria in the appendix.) To formalize this, let $ha$ denote the child node reached if the player plays action $a$ at node $h$. CDT postulates that if it plays according to $\boldsymbol{x}$, reached infoset $I$, and deviates this one time to action $a$, it anticipates to receive $\sum_{h \in I} \mathbb{P}(h \mid \boldsymbol{x}) \cdot U(\boldsymbol{x} \mid ha)$ utility from it overall. It can be shown that this quantity is equal to the partial derivative $\nabla_{I,a} U(\boldsymbol{x})$ of the utility function $U$ w.r.t. to action $a$ of infoset $I \in \mathcal{I}$ at $\boldsymbol{x}$ (Piccione & Rubinstein, 1997; Oesterheld & Conitzer, 2024).

**Definition 3.** *A strategy $\boldsymbol{x}$ is called an $\epsilon$-CDT equilibrium ($\epsilon \geq 0$) of a decision problem $\Gamma$ if for all infosets $I \in \mathcal{I}$ and all alternative randomized actions $\alpha \in \Delta(A_I)$, we have*

$$U(\boldsymbol{x}) \geq U_{\text{CDT}}(\alpha \mid \boldsymbol{x}, I) - \epsilon, \text{ where } U_{\text{CDT}}(\alpha \mid \boldsymbol{x}, I) := U(\boldsymbol{x}) + \sum_{a \in A_I} (\alpha(a) - \boldsymbol{x}(a \mid I)) \nabla_{I,a} U(\boldsymbol{x}).$$

Tewolde et al. (2023; 2024) observed that CDT equilibra correspond to *Karush-Kuhn-Tucker (KKT)* points, also known as first-order optima of constrained optimization, discussed further in Section 3.1.

---

**Algorithm 1:** A template for first-order optimization over products of simplices.

1   **Input:** Feasible set $\mathcal{X} = \Delta(m_1) \times \cdots \times \Delta(m_n)$, utility function $U : \mathcal{X} \to \mathbb{R}$
2   **for** $i = 1, \ldots, n$ **do**
3      Initialize local optimizer $\mathcal{R}_i$ on $\Delta(m_i)$
4      Set $\boldsymbol{u}_i^{(0)} \leftarrow \boldsymbol{0}$
5   **for** $t = 1, \ldots, T$ or *until convergence* **do**
6      **for** $i = 1, \ldots, n$ **do** $\tilde{\boldsymbol{u}}_i^{(t)} \leftarrow \boldsymbol{u}_i^{(t-1)}$ // *Set* $\tilde{\boldsymbol{u}}_i^{(t)} \leftarrow \boldsymbol{0}$ *instead if* $\mathcal{R}_i$ *is not* optimistic/predictive
7      **for** $i = 1, \ldots, n$ **do** $\boldsymbol{x}_i^{(t)} \leftarrow \mathcal{R}_i.\text{GETX}(\tilde{\boldsymbol{u}}_i^{(t)})$
8      **for** $i = 1, \ldots, n$ **do**
9         $\boldsymbol{u}_i^{(t)} \leftarrow \nabla_{\boldsymbol{x}_i} U(\boldsymbol{x}^{(t)})$
10        $\mathcal{R}_i.\text{STEP}(\boldsymbol{u}_i^{(t)})$
11   **return** $\boldsymbol{x}^{(t)}$

---

**Algorithm 2:** (Optimistic) Projected gradient descent; (O)GD

1   Initialize learning rate $\eta > 0$, $\hat{\boldsymbol{x}}^{(1)} \in \Delta(m)$
2   **procedure** GETX$(\tilde{\boldsymbol{u}}^{(t)})$   **return** $\boldsymbol{x}^{(t)} \leftarrow \Pi_{\Delta(m)}\big(\hat{\boldsymbol{x}}^{(t)} + \eta \tilde{\boldsymbol{u}}^{(t)}\big)$
3   **procedure** STEP$(\boldsymbol{u}^{(t)})$   $\hat{\boldsymbol{x}}^{(t+1)} \leftarrow \Pi_{\Delta(m)}\big(\hat{\boldsymbol{x}}^{(t)} + \eta \boldsymbol{u}^{(t)}\big)$

---

## 3   ALGORITHMS

This section dives into algorithmic approaches for tackling imperfect-recall decision problems. We will first review some known algorithms that will serve as our baselines in the experiments. In the second part, we introduce a family of algorithms from the game theory literature to the problem of nonlinear constrained optimization, which—as we shall see—performs remarkably well in practice.

### 3.1   KNOWN APPROACHES AND BASELINES

Despite the complexity barriers for computing optimal strategies (Proposition 2), one may still hope to come up with fast algorithms in practice. For that reason, we make use of a popular commercial solver for nonlinear optimization, Gurobi [2025], which guarantees global optimality (up to a small tolerance error) upon termination. A description of our implementation can be found in the appendix. We will see that this approach scales poorly in our benchmarks.

This motivates shifting our attention to CDT equilibria, which—as we mentioned—can be expressed as KKT points of a polynomial optimization problem. It is well known that $\epsilon$-KKT points can be computed in $\text{poly}(1/\epsilon)$ time via *(projected) gradient descent (GD)* (for example, Fearnley et al., 2023). This will serve as our basic benchmark when it comes to algorithms for computing CDT equilibria. We will also experiment with the following two popular variants of GD: (1) *Optimistic (projected) gradient descent (OGD)*, which goes back to Popov (1980), and is receiving renewed interest in recent years, especially in the context of games (Wei et al., 2021; Daskalakis & Panageas, 2018; Daskalakis et al., 2018). And (2) *AMSGrad (AMS)* (Reddi et al., 2018), an adaptive gradient method based on exponential moving averages for the first and second gradient momentum that also enjoys theoretical convergence guarantees.

Since we deal exclusively with optimization over a product of simplices, we can provide a basic template for decomposing it into independent subproblems over the individual simplices (in Algorithm 1). What remains to be specified is the choice of individual local optimizers. Algorithm 2, for example, describes (O)GD. We present the AMS algorithm in the appendix, together with an explanation on how to implement its projection operator in simplex domains. Regarding implementing these algorithms and the upcoming RM ones, a non-trivial observation is that, for tree-form decision problems, the gradients at every decision point can be computed in total time linear in the size of the decision problem. Indeed, the quantities $\mathbb{P}(h \mid \boldsymbol{x}^{(t)})$ and $U(\boldsymbol{x}^{(t)} \mid ha)$ in CDT utilities (Section 2.2) can be computed for each history $h$ by recursive passes down and up through the tree respectively.

| **Algorithm 3:** (Pred.) Reg. matching; $(\mathrm{P})\mathrm{RM}$ | **Algorithm 4:** $(\mathrm{P})\mathrm{RM}^+$ |
|---|---|
| 1 Initialize $\boldsymbol{r}^{(1)} \leftarrow \boldsymbol{0}, \boldsymbol{x}^{(0)} \in \Delta(m)$ | 1 Initialize $\boldsymbol{r}^{(1)} \leftarrow \boldsymbol{0}, \boldsymbol{x}^{(0)} \in \Delta(m)$ |
| 2 **procedure** $\mathrm{GETX}(\tilde{\boldsymbol{u}}^{(t)})$ | 2 **procedure** $\mathrm{GETX}(\tilde{\boldsymbol{u}}^{(t)})$ |
| 3 $\quad \boldsymbol{\theta}^{(t)} \leftarrow \big[\boldsymbol{r}^{(t)} + \tilde{\boldsymbol{u}}^{(t)} - \big\langle \tilde{\boldsymbol{u}}^{(t)}, \boldsymbol{x}^{(t-1)} \big\rangle \boldsymbol{1}\big]^+$ | 3 $\quad \boldsymbol{\theta}^{(t)} \leftarrow \big[\boldsymbol{r}^{(t)} + \tilde{\boldsymbol{u}}^{(t)} - \big\langle \tilde{\boldsymbol{u}}^{(t)}, \boldsymbol{x}^{(t-1)} \big\rangle \boldsymbol{1}\big]^+$ |
| 4 $\quad$ **if** $\boldsymbol{\theta}^{(t)} \neq \boldsymbol{0}$ **then** $\boldsymbol{x}^{(t)} \leftarrow \boldsymbol{\theta}^{(t)} / \big\|\boldsymbol{\theta}^{(t)}\big\|_1$ | 4 $\quad$ **if** $\boldsymbol{\theta}^{(t)} \neq \boldsymbol{0}$ **then** $\boldsymbol{x}^{(t)} \leftarrow \boldsymbol{\theta}^{(t)} / \big\|\boldsymbol{\theta}^{(t)}\big\|_1$ |
| 5 $\quad$ **else** $\boldsymbol{x}^{(t)} \leftarrow \boldsymbol{x}^{(t-1)}$ | 5 $\quad$ **else** $\boldsymbol{x}^{(t)} \leftarrow \boldsymbol{x}^{(t-1)}$ |
| 6 $\quad$ **return** $\boldsymbol{x}^{(t)}$ | 6 $\quad$ **return** $\boldsymbol{x}^{(t)}$ |
| 7 **procedure** $\mathrm{STEP}(\boldsymbol{u}^{(t)})$ | 7 **procedure** $\mathrm{STEP}(\boldsymbol{u}^{(t)})$ |
| 8 $\quad \boldsymbol{r}^{(t+1)} \leftarrow \boldsymbol{r}^{(t)} + \boldsymbol{u}^{(t)} - \big\langle \boldsymbol{u}^{(t)}, \boldsymbol{x}^{(t)} \big\rangle \boldsymbol{1}$ | 8 $\quad \boldsymbol{r}^{(t+1)} \leftarrow [\boldsymbol{r}^{(t)} + \boldsymbol{u}^{(t)} - \big\langle \boldsymbol{u}^{(t)}, \boldsymbol{x}^{(t)} \big\rangle \boldsymbol{1}]^+$ |

## 3.2 Regret matching for constrained optimization

We now introduce a new family of algorithms for constrained optimization based on *regret matching (RM)* (Hart & Mas-Colell, 2000) (Algorithm 3). We use the notation $[\boldsymbol{x}]^+ := \max(\boldsymbol{x}, \boldsymbol{0})$ for a vector $\boldsymbol{x} \in \mathbb{R}^m$, and $\boldsymbol{1}$ for the all-ones vector. $\mathrm{RM}^+$ is a simple variant of $\mathrm{RM}$ that has been shown to work very well in practice (*e.g.*, Bowling et al., 2015); the only difference is that $\mathrm{RM}^+$ truncates the regrets in each iteration (Algorithm 4). We also implement their predictive versions $\mathrm{PRM}$ and $\mathrm{PRM}^+$ (Farina et al., 2021). All these algorithms are designed to minimize regret in the online learning setting. In perfect-recall zero-sum games, having vanishing regret implies that the *average* strategies converge to the set of Nash equilibria, whereas the last iterate can fail to converge (Farina et al., 2023).

Although $\mathrm{RM}$ and its variants have received a lot of attention for perfect-recall zero-sum games, there was hitherto little reason to believe they would perform well in constrained optimization problems.[1] Since $\mathrm{RM}$ is a regret minimizer (Hart & Mas-Colell, 2000), we note that $\mathrm{RM}$ reaches global optimality in concave settings. A formal proof is in the appendix.

**Proposition 4.** *If the objective function $U$ is a concave polynomial over $\mathcal{X}$, then $\mathrm{RM}$ converges in best iterate to the global maximum of $U$ over $\mathcal{X}$.*

Unlike for gradient descent methods, however, it is not known whether $\mathrm{RM}$ variants converge to first-order optima for generic nonconcave maximization problems such as ours.

## 4 Benchmarks

We introduce three different parametric classes of decision problems. The parameters dictate the structure of the problem instance such as its depth, number of infosets, the degree of absentmindedness, and number of actions per infoset, etc. Our implementation is based on LiteEFG (Liu et al., 2024), a lightweight format for extensive-form games.

## 4.1 Simulation problems

Inspired by the type of problems discussed in the introduction, we model problems that involve simulating an agent. For this to be effective, the simulation must be indistinguishable from reality; thus, nodes corresponding to decisions in simulation are in the same infoset as nodes corresponding to decisions in reality. Specifically, we consider games where in the simulation phase, the simulator may test the simulated agent's behavior, possibly multiple times in a row. The agent will then be deployed if and only if it acted as intended in simulation.

A concrete yet simple simulation example is given in Figure 2 (left). In line with previous works on simulation games, we focus on the setting in which the agent has only two actions: "good" or "bad" (with respect to the simulator's goals). If the agent ever acts bad in simulation, the game ends and the agent receives some constant utility (0, by default). We fix the simulator's strategy, thus making the simulator a chance node and this game a single-agent problem. The simulator can simulate the

---

[1] We discuss in the appendix why the CFR framework cannot be applied to our setting, and we distinguish our $\mathrm{RM}$ methods from CFR.

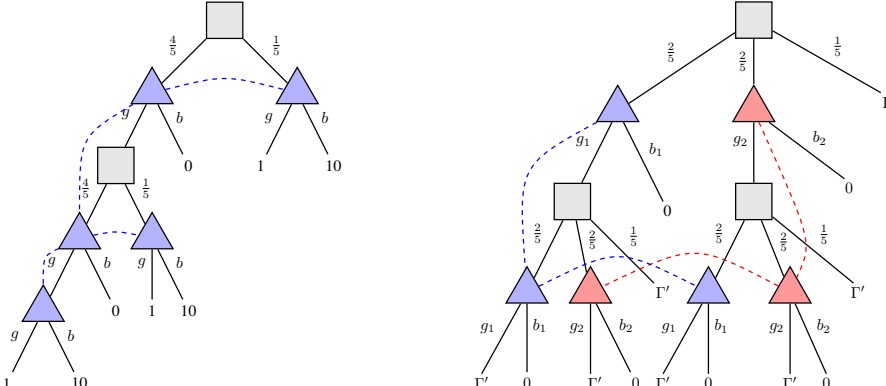

Figure 2: Left: A simple simulation problem. The agent is misaligned: it receives 10 utility for its preferred action (which the simulator considers the bad action), and 1 utility for the other action. The simulator decides to simulate the agent with the fixed probability $4/5$, and up to a total of 2 times, in order to catch misaligned behavior in advance. Right: A slightly more complex simulation problem. There are now two scenarios the agent can be tested on, giving rise to two different infosets. Reaching the subtree $\Gamma'$ means the agent reached deployment; we visualize $\Gamma'$ in the appendix.

agent up to $n$ times, but does not have to; whether they simulate the agent yet another time will be decided by a probability parameter. Once the agent reaches a decision node in the deployment phase, it will receive utility $\gamma \in \mathbb{R}$ for acting good and $\beta \in \mathbb{R}$ for acting bad (specified further below).

The purpose of simulating the agent, and thus inducing absentmindedness, is two-fold (Chen et al., 2024). First, it allows the simulator to *screen* for misaligned agents: if the agent acts bad with some positive probability, it becomes exponentially unlikely—in the number of simulation rounds—to remain unnoticed by the simulator. In contrast, if the agent prefers to act good ($\gamma > \beta$), it chooses to act so deterministically, guaranteeing that it will reach the deployment phase. Second, the simulations have a *disciplining* effect: even in the worst case where the simulator is presented with a misaligned agent ($\gamma < \beta$), the simulator still incentivizes the agent to act good most of the time, if not all the time, by testing the agent (multiple times) in simulation.

We expand on prior work by allowing the simulator to evaluate the agent in $k \geq 1$ *scenarios* rather than just a single one. For example, an autonomous vehicle might be tested on its behavior in the city, on the highway, off-road, and under certain difficult weather conditions. A language model might be evaluated on writing essays and executable code, and providing mental support through conversation. Furthermore, we also extend the deployment phase to $m \geq 1$ rounds, with acting good and bad in scenario $i$ contributing with $\beta_i$ and $\gamma_i$ to the total payoffs. An example of such a simulation problem is given in Figure 2 (right). In particular, an agent might now refrain from ever acting bad in scenario $i$ because it hopes to act bad (or act at all) in another scenario $i'$ in deployment.

## 4.2 SUBGROUP DETECTION UNDER PRIVACY CONSTRAINTS

Motivated by the privacy applications discussed in the introduction, we introduce a parametrized class of decision problems in which the agent aims to identify *suitable* candidates—be it medical patients, investment opportunities, and so on—under privacy constraints. The agent is provided with a connection graph, such as in Figure 3 (left), which captures some notion of relationship or similarity between the candidates (possibly based on social ties, physical characteristics, geography, etc.). Connected subsets of suitable candidates, called *subgroups*, are present in the graph unknown to the agent. An action in the corresponding decision problem consists of selecting one of the candidates, and the goal is to maximize the number of distinct candidates selected from the suitable subgroups. If a selected candidate is a member of a subgroup, then the agent observes this fact henceforth; otherwise, the agent forgets having chosen this candidate at all (since it is sensitive info to know the unsuitable ones). Figure 3 (right) depicts an example; it resembles the prominent "Battleship" game, except that here the agent is absentminded about cells selected in the past that did not hit a ship. The parameters in this benchmark problem class control (a) the number of rounds

the agent can select candidates for; (b) an underlying graph structure, such as a 2D grid, or Erdős-Rényi $G(n, p)$ and $G(n, m)$ graph models; (c) the subgroups in the graph in terms of quantity, sizes, shapes (lines, cycles, cliques, stars) as well as the procedure for distributing them in the graph; and (d) the immediate payoffs for selecting suitable candidates of different subgroups.

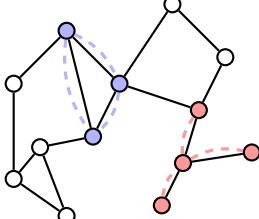 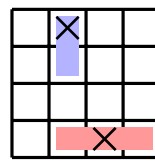

Figure 3: Subgroup detection under privacy constraints. On the left, we see an arbitrary graph with two subgroups (a 3-clique, and a star of degree 3). The goal is to find as many of the subgroups' nodes as possible. On the right, we see another such decision problem on a 2D grid, which we visualized as an instance of the Absentminded Battleship game. The agent has already succeeded in hitting one node of each ship, which indicates that there must be more subgroup nodes nearby. The agent does not remember whether it has selected any cell other than these two before.

### 4.3 RANDOM DECISION PROBLEMS

Finally, we introduce a highly parametrized class of randomly generated decision problems, following an active line of work on random games (McKelvey & McLennan, 1996; Nudelman et al., 2004; Arieli & Babichenko, 2016; Amiet et al., 2021; Flesch et al., 2023; Heinrich et al., 2023). Part of their appeal is that they serve as a sanity check and help counterbalance cherry picking of benchmark problems. The parameters dictate (a) the probability with which a node will be terminal (dependent on its depth), (b) the probabilities with which a nonterminal node has $k$ available actions, as well as with which it will be a chance node, (c) the (approximate) number of nodes we want to cover with each infoset, and (d) the probability distribution over payoffs at terminal nodes. The payoffs at the leaf nodes are drawn uniformly at random between $0$ and $1$. In the experiments in Section 5, each tree has varying depth in an interval $[d, d']$ where $4 \leq d \leq d' \leq 15$, the nonterminal nodes have 3 to 5 available actions and a 20% probability to be a chance node, and infosets are of a size roughly proportional to $n^{2/3}$, where $n$ denotes the total number of decision nodes in the tree.

## 5 EXPERIMENTAL EVALUATION

Having introduced our benchmarks, we now use them to evaluate the performance of the algorithms described earlier in Section 3. Abbreviations "Sim," "Det," and "Rand" stand for simulation problems (Section 4.1), subgroup detection problems (Section 4.2), and random problems (Section 4.3) respectively. The suffixes indicate the number of nodes in the decision tree (with "k" and "m" abbreviating thousands and millions). Our algorithms run until any of three termination conditions is met: achieving a KKT gap of at most $10^{-6}$, reaching the time limit of $4$ hours,[2] or reaching the iteration limit of 6000. We run the first-order methods for 12 times with randomly initialized strategies and report the median. For GD and OGD, we run the algorithm with different learning rates, namely $\eta \in \{1, 10^{-1}, 10^{-2}, 10^{-3}\}$, and report only the one that minimizes the KKT gap the fastest at time of termination. We operate analogously for AMS, except that we instead test the parameter settings $(\eta, \beta_1, \beta_2) \in \{10^{-1}, 10^{-2}\} \times \{0.9, 0.99\} \times \{0.99, 0.999\}$ (this includes Reddi et al.'s suggested $\beta$-values). A subset of our results are gathered in Table 1 and aggregate statistics in Table 2. We also plot the KKT gap and value versus iteration in Figure 4 to gain insight into the process of convergence.[3] The shaded regions represent the best and worst run for the respective iteration count and across the 12 initializations. Further experimental details and results can be found in the appendix.

---

[2]With the only exceptions of Det-{9m,10m,18m} problems, which we run for 12 hours since the standard time limit poses a significant bottleneck for those instances.

[3]Our regret matching implementations complete more iterations per time than our gradient descent implementations, so the fact that we plot against iterations rather than time favors the gradient descent algorithms.

| Problem | Gurobi value | time | GD value | time | gap | OGD value | time | gap | AMS value | time | gap | RM value | time | gap | RM$^+$ value | time | gap | PRM$^+$ value | time | gap |
|---|---|---|---|---|---|---|---|---|---|---|---|---|---|---|---|---|---|---|---|---|---|
| Det-1k | **13.00** | 1m 24s | **13.00** | 0.13s | — | **13.00** | 0.07s | — | **13.00** | 1.04s | — | **13.00** | 0.32s | — | **13.00** | 0.36s | — | **13.00** | 0.41s | — |
| Det-1.8k | **22.00** | 2m 40s | **22.00** | 0.06s | — | **22.00** | 0.07s | — | **22.00** | 0.71s | — | **22.00** | **0.03s** | — | **22.00** | 0.03s | — | **22.00** | 0.03s | — |
| Det-2.0k | **17.50** | 1m 42s | **17.50** | 0.03s | — | **17.50** | 0.05s | — | **17.50** | 0.20s | — | **17.50** | **0.03s** | — | **17.50** | **0.03s** | — | **17.50** | **0.03s** | — |
| Det-2.1m | — | — | 26.00 | — | 1e-05 | 25.96 | — | 0.02 | **26.15** | — | 0.002 | **26.15** | — | 0.003 | **26.15** | 3h 25m | — | **26.15** | — | 0.005 |
| Det-2.2m | — | — | 16.20 | — | 0.002 | 15.93 | — | 0.02 | 16.36 | — | 0.0002 | 16.36 | **2h 22m** | — | 16.36 | 3h 13m | — | **16.36** | — | 5e-06 |
| Det-3.8m | — | — | 15.66 | — | 0.003 | 15.14 | — | 0.03 | 15.78 | — | 0.0002 | **15.80** | — | 2e-06 | **15.80** | — | 5e-05 | **15.80** | — | 0.0003 |
| Det-9m | — | — | 23.16 | — | 0.004 | 22.71 | — | 0.02 | **23.45** | — | 0.004 | **23.45** | — | 0.0001 | **23.45** | — | 0.0001 | **23.45** | — | 0.0004 |
| Det-10m | — | — | 24.64 | — | 0.002 | 24.61 | — | 0.003 | 24.76 | — | 0.009 | 24.76 | — | 0.002 | **24.76** | — | **0.0004** | 24.76 | — | 0.0008 |
| Det-18m | — | — | 26.38 | — | 0.006 | 25.81 | — | 0.05 | **26.71** | — | 0.004 | 26.71 | — | 0.004 | **26.71** | — | **0.001** | 26.71 | — | 0.04 |
| Rand-24k | **0.72** | — | 0.66 | 7m 0s | — | 0.66 | 7m 46s | — | 0.66 | 4m 4s | — | 0.66 | **26.55s** | — | 0.66 | 1m 3s | — | 0.66 | 5m 5s | — |
| Rand-35k | **1.00** | — | 0.95 | 3.85s | — | 0.95 | 3.76s | — | 0.95 | 3.90s | — | 0.92 | **0.99s** | — | 0.92 | 1.18s | — | 0.94 | 1.68s | — |
| Rand-42k | **0.69** | — | 0.55 | — | 0.01 | 0.55 | — | 0.01 | 0.64 | — | 0.0006 | 0.65 | — | 2e-06 | 0.65 | 5m 56s | — | 0.65 | **3m 19s** | — |
| Rand-13m | — | — | 0.59 | — | 0.003 | 0.58 | — | 0.003 | **0.65** | 1h 40m | — | 0.63 | 19m 11s | — | 0.64 | **17m 31s** | — | **0.65** | 36m 42s | — |
| Rand-18m | — | — | 0.97 | 2h 33m | — | 0.97 | 3h 0m | — | **0.99** | 1h 29m | — | 0.95 | 29m 45s | — | 0.97 | 24m 0s | — | 0.97 | **14m 31s** | — |
| Rand-23m | — | — | 0.94 | 3h 37m | — | 0.93 | — | 0.0007 | **0.98** | 3h 20m | — | **0.98** | 23m 10s | — | 0.96 | 23m 5s | — | 0.95 | **18m 2s** | — |
| Sim-3k | **6.25** | 1m 1s | **6.25** | 0.32s | — | **6.25** | 1.03s | — | **6.25** | 5.54s | — | **6.25** | **0.26s** | — | **6.25** | 0.28s | — | **6.25** | 0.48s | — |
| Sim-7k | **8.58** | 1m 36s | **8.58** | 0.05s | — | **8.58** | 0.05s | — | **8.58** | 0.12s | — | **8.58** | **0.05s** | — | **8.58** | **0.05s** | — | **8.58** | 0.05s | — |
| Sim-13k | **10.38** | 4m 21s | **10.38** | **0.69s** | — | **10.38** | 8.54s | — | **10.38** | 14.37s | — | **10.38** | 1.03s | — | **10.38** | 1.01s | — | **10.38** | 3.97s | — |
| Sim-540k | 6.41 | — | **8.54** | 47.54s | — | **8.54** | 2m 37s | — | **8.54** | 15m 31s | — | **8.54** | 19.39s | — | **8.54** | 19.44s | — | **8.54** | 3m 3s | — |
| Sim-1m | 4.14 | — | **4.77** | 5m 33s | — | **4.77** | 7m 2s | — | **4.77** | 29m 22s | — | **4.77** | **2m 14s** | — | **4.77** | 2m 34s | — | **4.77** | 4m 20s | — |
| Sim-1.9m | — | — | **13.45** | 18.31s | — | **13.45** | 17.96s | — | **13.45** | 1m 2s | — | **13.45** | 12.36s | — | **13.45** | **12.19s** | — | **13.45** | 12.47s | — |
| Sim-2.3m | — | — | **11.09** | 22.01s | — | **11.09** | 21.88s | — | **11.09** | 1m 10s | — | **11.09** | **14.97s** | — | **11.09** | 15.00s | — | **11.09** | 15.13s | — |
| Sim-4m | — | — | **14.01** | 45m 5s | — | **14.01** | 41m 0s | — | 13.98 | — | 0.02 | **14.01** | 11m 36s | — | **14.01** | **7m 3s** | — | **14.01** | 21m 17s | — |

Table 1: The performance of various algorithms in a subset of our benchmarks. Value and convergence winners per game highlighted in bold. For `Gurobi`, time is only reported if it terminated.

| Category | Metric | Gurobi | GD | OGD | AMS | RM | RM$^+$ | PRM | PRM$^+$ |
|---|---|---|---|---|---|---|---|---|---|
| Utility value | % of best (↑) | 36.1 | 42.6 | 42.6 | **78.7** | 65.6 | 70.5 | 72.1 | 72.1 |
| | avg. rank (↓) | 6.26 | 5.12 | 5.34 | **3.60** | 4.04 | 3.83 | 3.89 | 3.91 |
| Convergence | % reached (↑) | 27.9 | 72.1 | 72.1 | 77.0 | 83.6 | **90.2** | 78.7 | 80.3 |
| | % of best (↑) | 0.0 | 11.5 | 6.6 | 0.0 | **41.0** | 39.3 | 23.0 | 21.3 |
| | avg. rank (↓) | 8.00 | 4.80 | 5.52 | 5.74 | 2.34 | **2.17** | 3.84 | 3.59 |

Table 2: Aggregate statistics of the full experiments in the appendix. We report how often an algorithm reached convergence across all 61 benchmark instances, and relative to the other algorithms, how well it ranks on average (lower is better), and how often it achieved the best value / convergence.

The main takeaways are the following:

1. `Gurobi` fails to converge beyond small instances ($\leq$100k nodes for simulation, and $\leq$20k otherwise). Moreover, when it reaches termination, the time required is multiple orders of magnitude more than that of the first-order optimizers. This is despite the fact that `Gurobi` is based on an optimized C++ implementation whereas our first-order optimizers are implemented in Python.
2. Interestingly, in all such cases in Table 1, where we know the optimal value, the first-order optimizers converge to an optimal strategy.

As expected, we can also find some experiments where the previous point will not hold (*e.g.* Rand-42k once `Gurobi` would eventually terminate). Indeed, we design an extreme example in the appendix where our gradient descent and regret matching methods all converge to a KKT point that is arbitrarily bad in value relative to the global optimum. We also illustrate on two examples that one of the two algorithm types can perform arbitrarily bad while the other reaches global optimum.

3. `AMS` and `RM`$^+$ oftentimes attain higher values than `GD` and `OGD`, and almost never less.
4. The `RM` family of algorithms, and `RM`$^+$ in particular, consistently outperform `GD`, `OGD`, and `AMS` in runtime. The difference is often many orders of magnitude, especially in the larger instances.

The literature on two-player zero-sum *perfect-recall* games may offer some conceptual explanation here. Namely, it is believed that the following property of RM methods—which continues to be satisfied in our setting with imperfect recall—is important to practical performance: at the infoset level, RM methods are invariant to step sizes (Chakrabarti et al., 2024). Instead, they regulate their step sizes based on the current and past gradients alone, requiring no careful step size tuning in the same way that gradient descent does.

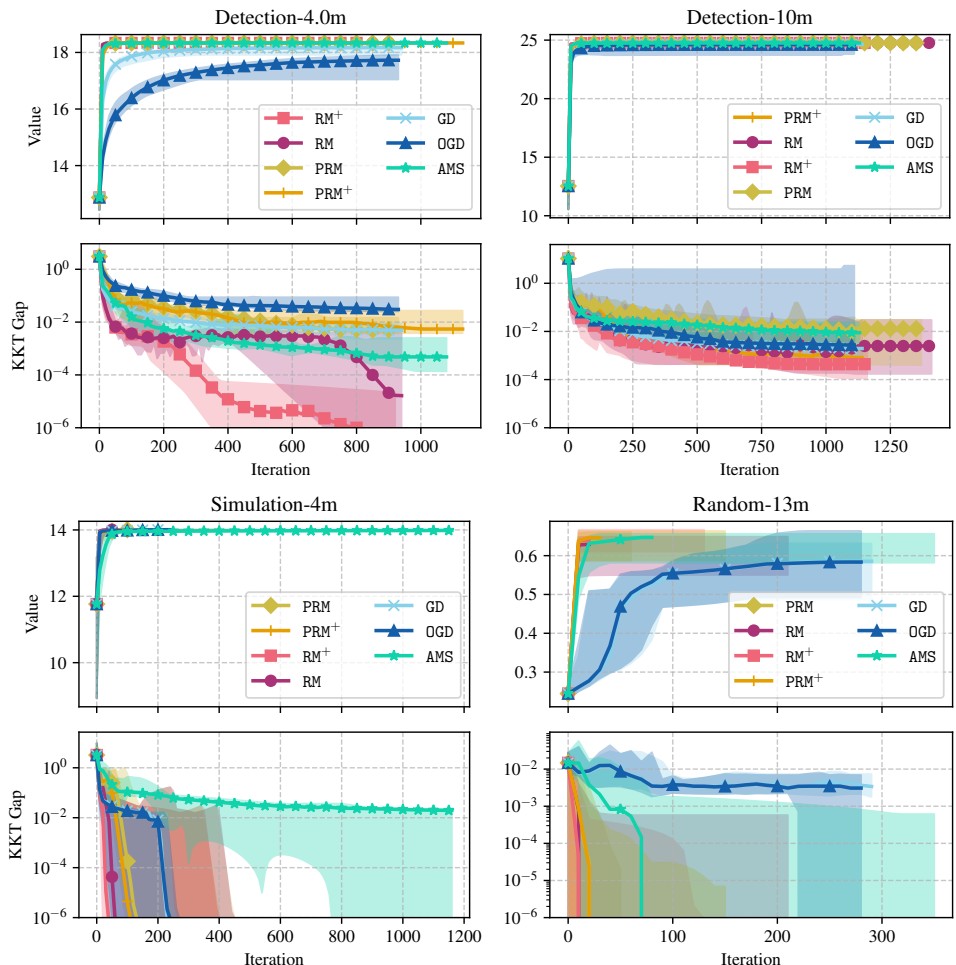

Figure 4: Two detection instances, a simulation and a random instance. They have ∼1200 infosets, ∼300 infosets, 3 infosets, and ∼100 infosets respectively.

5. $\text{RM}^+$ performs best among the RM family. Surprisingly, it typically outperforms $\text{PRM}^+$, which stands in stark contrast to what has been observed in zero-sum games (Farina et al., 2021).

Our intuition for that is as follows. Predictiveness in RM (= Optimism in GD) roughly corresponds to having negative momentum, which is beneficial in zero-sum games and minimax optimization because it helps minimize regret faster. But in our setting of nonlinear (single-player) optimization, it is not known whether predictiveness helps anymore, since the task is not to minimize regret but to search for a first-order optimal point. Indeed, our experiments seem to suggest otherwise.

## 6 FUTURE RESEARCH

Our paper opens many interesting avenues for future work. First, we have focused exclusively on solving tabular imperfect-recall decision problems. A promising direction is to use modern RL techniques to expand the scope to even larger problems that cannot be represented in tabular form. Considering other formulations beyond tree-form decision problems, such as (PO)MDPs, is another natural direction that was beyond our scope. Finally, our experiments revealed that the regret matching family of algorithms is a formidable first-order optimizer; elucidating their theoretical properties is another important open question.

## REPRODUCIBILITY STATEMENT

Comprehensive details about our experimental methodology can be found in Sections 4 and 5 as well as in the last section of the appendix, including generation procedures, hyperparameter ranges, termination conditions, hyperparameter grids, and hardware specifications. The supplementary material contains our code base for generating instances from our benchmark suite, running the discussed algorithms on problem instances, and evaluating and visualizing the results. The specific 61 benchmark instances used in our evaluation, along with their corresponding experimental results, are available through the provided link at the end of the appendix.

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

## A APPENDIX

Here, we expand further on details we omitted in the main body.

### A.1 FURTHER RELATED WORK

**Simulation games** Commencing from the paper of Kovařík et al. (2023), there has been significant interest in situations where one player can simulate another player (Chen et al., 2024; Kovařík et al., 2024; 2025b; Oesterheld, 2019; Cooper et al., 2025); this is precisely the type of problem captured by one of our benchmarks. The premise of simulating the other player is strongly connected with the notion of *program equilibrium* (Tennenholtz, 2004), where players are allowed to submit source code. This turns out to unlock more cooperative outcomes by expanding the set of equilibria.

**MDPs and repeated games** Another notable motivation for examining imperfect-recall decision problems lies in the fact that they can result in simpler and more interpretable strategies. This point can be illustrated well in the context of Markov decision problems (MDPs), where insisting on *Markovian* policies—which depend solely on the state and not the entire history—is particularly common; this can be viewed as an extreme form of imperfect recall. Relatedly, restricting the memory and description complexity of a policy has received a lot of attention in the context of repeated games (*e.g.*, Foster & Hart, 2018; Papadimitriou & Yannakakis, 1994). In certain settings, near-optimal policies are possible even under imperfect recall. More broadly, the question of characterizing the value of recall was recently addressed by Berker et al. (2025).

**CDT equilibria** The CDT equilibrium falls under the family based on the *multi-selves* approach (Kuhn, 1953). At a high level, whenever the player has imperfect information on the decision node of an infoset it is currently in, the player will weight each possibility with the probability of reaching the decision node in question under strategy $x$. The name is derived from the intuition that the player's choice to deviate from $x$ at the current node does not *cause* any change in its behavior at any other node, even if they are of the same infoset. Another prominent member is the *EDT equilibrium*, which results from marrying evidential decision theory with the multi-selves approach (Oesterheld & Conitzer, 2024). For further background on the ongoing debate around decision theories and how they relate to belief formation (*cf.* the "sleeping beauty" problem (Elga, 2000)), we refer to (Piccione & Rubinstein, 1997; Briggs, 2010; Oesterheld & Conitzer, 2024). Further, we refer to Tewolde et al. (2024) for a computational treatment of equilibria in multi-player games with imperfect recall. With regard to the complexity of computing CDT equilibria, we saw earlier that a $\text{poly}(1/\epsilon)$ time algorithm exists by running GD on a suitable optimization problem; in the regime where $\epsilon$ is exponentially small, the complexity is characterized by the class CLS, and is believed to be hard (Daskalakis & Papadimitriou, 2011; Fearnley et al., 2023; Tewolde et al., 2023). Conceptually, and also computationally, CDT equilibria in decision problems with imperfect recall have been also connected to Nash equilibria in team games that respect a given set of game symmetries (Lambert et al., 2019; Emmons et al., 2022; Tewolde et al., 2025).

**Regret matching** Regret matching and its variants have received a lot of attention in (two-player) zero-sum *perfect recall* extensive-form games. In particular, the *counterfactual regret minimization (CFR)* algorithm, famously introduced by Zinkevich et al. (2007), employs a separate RM algorithm for each information set. The CFR framework has spawned a flourishing, and still active, line of work. Yet, much less is known beyond (two-player) zero-sum games. It has to be stressed again that in zero-sum games, RM and its variants only have guarantees concerning the time average strategy. In fact, the last iterate can fail to converge (Farina et al., 2023). Our experiments suggest a fundamental difference in constrained optimization problems: all our results make use of the last iterate, which not only converges, but does so remarkably fast. To our knowledge, there is currently no theory that predicts that RM and its variants will converge. The continuous time of RM was analyzed by Hart & Mas-Colell (2003), who also established asymptotic convergence in two-player potential games for a certain—somewhat artificial—variant of RM in discrete time. Fast empirical convergence was reported by Ma & Gerber (2014) in a certain class of congestion games.

An intriguing behavior we uncover in this paper is that the RM family of algorithms often outperforms (O)GD in terms of the attained value, at least for the benchmark problems we consider. In the context of multi-player potential games, which is closely related to imperfect-recall decision problems, the

problem of characterizing the performance of different algorithms is poorly understood. One notable contribution here is the recent paper of Sakos et al. (2024), but it only focused on $2 \times 2$ games. Providing a theoretical explanation that justifies the excellent performance of RM in terms of value is an interesting but challenging direction for the future.

**Game abstractions and related work**    As stated in the introduction, games with imperfect recall have found great success in the state-of-the-art algorithms for solving real-world games that are too large to handle, and therefore need to be compressed in a game abstraction (Waugh et al., 2009; Čermák et al., 2017a; Brown & Sandholm, 2018; 2019; Benjamin & Lanctot, 2024; Li & Huang, 2025). The motivation and techniques from that line of literature are complementary to ours because

1. they can flexibly change the forms and structures of imperfect recall since the there is an underlying *perfect-recall* game that the user cares about and an imperfect-recall abstraction is evaluated by how well its solutions can be lifted to good strategies in the underlying perfect-recall game, and because

2. their forms of imperfect recall are *designed* to be relatively benign so that the solutions in the imperfect-recall abstraction can be computed efficiently (recall the hardness of Proposition 2).

Our work, on the other hand, is interested in decision problems that inherently exhibit imperfect recall, and solving them for their own sake (there is no underlying perfect-recall problem), even if they are computationally hard (such as when absentminded infosets are present).

Various narrower forms of imperfect recall have received wide attention. Decision problems without absentmindedness always admit optimal solutions in pure strategies, which enables methods based on mixed-integer programming and double oracle-style incremental strategy generation (Čermák et al., 2017a;b; 2020). Prominent forms of imperfect recall that are computationally benign include skew well-formed games, for which CFR is still guaranteed to work (Lanctot et al., 2012; Kroer & Sandholm, 2016), and decision problems with A-loss recall (Kaneko & Kline, 1995; Kline, 2002), for which optimal strategies can be computed in polynomial time (Čermák et al., 2018; Gimbert et al., 2025). CFR methods have also shown experimental success in slightly more general settings of imperfect-recall abstractions (Kroer & Sandholm, 2016; Čermák et al., 2020; Li & Huang, 2025), though these settings continue to be narrow subclasses of decision problems *without absentmindedness*. Indeed, CFR is a framework designed for the perfect recall setting, and past work—such as Waugh et al. (2009, Section "Challenges of Imperfect Recall")—have discussed why CFR conceptually cannot be extended to imperfect-recall settings beyond narrow subclasses such as (variants of) skew well-formed games. CFR updates the action probabilities at each infoset based on a notion of expected utilities that counterfactually assumes that the player played in the past as if it only wanted to reach the infoset in question. Under imperfect recall, and especially absentmindedness, notions such as "in the past" and "playing actions in order to reach an infoset" (once, or multiple times?) become highly dubious. In order to apply RM methods on decision problems with arbitrary forms of imperfect recall, we step away from CFR in this paper, and instead work in the *agent-form* of the decision problem Kuhn (1953). The agent-form imagines each infoset as being played by a separate player, therefore introducing multi-agent equilibrium concepts even in single-agent decision problems. Moreover, we tackle infosets with absentmindedness through causal decision theory (and a belief formation system that is compatible with it; *cf.* Tewolde et al., 2024). This equates to regret matching methods being applied on gradients from the polynomial optimization problem Equation (1). Last but not least, we are not studying RM methods as regret minimizers—as it is the case in the literature on imperfect-recall abstractions, using the CFR framework—but instead evaluate RM methods in their performance as a first-order optimizers. These two are incomparable.

**Mixed strategies and team games**    Much of the prior work in extensive-form games has focused on *mixed* strategies—probability distributions over pure strategies. Unlike behavioral strategies, mixed strategies allow the player to correlate its actions across infosets; one such example is *ex-ante* team coordination (Farina et al., 2018) in the context of team games. As we explained in our introduction, a team game can be phrased as an imperfect-recall decision problem; in fact, one without absentmindedness. Without absentmindedness, it follows that there exists an optimal strategy that is pure; in contrast, the presence of absentmindedness—which is primary focus on this paper—requires randomization (Isbell, 1957). In the presence of imperfect recall, mixed strategies are not

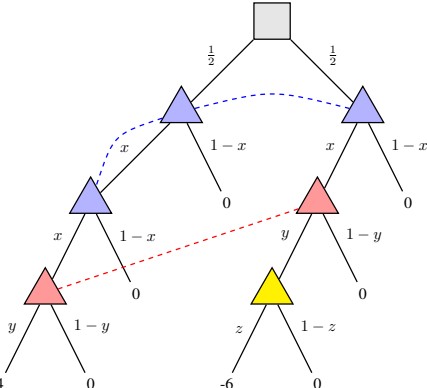

Figure 5: This is the decision making instance one would obtain from applying the construction of the proof of Theorem 1 to the polynomial maximization $\max 2x^2 y - 3xyz$ s.t. $0 \leq x, y, z \leq 1$.

realization-equivalent to behavioral strategies (Kuhn, 1953), and they do not fit our motivation since they imply a form of memory mechanism. Related to *ex-ante* team coordination, classical equilibrium concepts in extensive-form games involving *correlation* can be modeled via a mediator—a trusted third party—with imperfect recall (Zhang & Sandholm, 2022); that the mediator has imperfect recall can serve to safeguard the players' sensitive information, which is tied to one of the key motivations of this paper.

## A.2 ON THEOREM 1: AN EQUIVALENCE TO CONSTRAINED POLYNOMIAL OPTIMIZATION

Starting with part 1 of Theorem 1, the concrete polynomial optimization problem for maximizing utility in a decision making instance takes the following form:

$$
\begin{aligned}
\max_{\boldsymbol{x} \in \times_{I \in \mathcal{I}} \mathbb{R}^{A_I}} \quad & U(\boldsymbol{x}) = \sum_{z \in \mathcal{Z}} u(z) \cdot \mathbb{P}(z \mid \boldsymbol{x}) \\
\text{s.t.} \quad & \boldsymbol{x}(a \mid I) \geq 0 \quad \forall I \in \mathcal{I}, \forall a \in A_I \\
& \sum_{a \in A_I} \boldsymbol{x}(a \mid I) = 1 \quad \forall I \in \mathcal{I}
\end{aligned}
\tag{1}
$$

Recall that $\mathbb{P}(z \mid \boldsymbol{x})$ is a product of action probabilities $\boldsymbol{x}(a \mid I)$ and chance probabilities of actions the agent and chance need to take in order to reach leaf node $z$. The former kind are the optimization variables in the above program, and the latter kind are fixed scalars.

Next, we move our attention to part 2 of Theorem 1. As an example, take the polynomial maximization instance

$$
\max 2x^2 y - 3xyz \quad \text{s.t.} \quad 0 \leq x, y, z \leq 1 \,.
$$

Figure 5 then depicts its corresponding decision making instance under imperfect recall that one obtains from the construction in the proof of Theorem 1. The more general construction idea is as follows. Variables correspond to infosets, and occurrences of a variable to a decision node. For a hypercube domain, each decision node will have two actions; for general products of simplices, the number of actions at a decision node associated to a variable $x$ is precisely the number of vertices in the simplex that constrains $x$. A chance node at the root selects uniform randomly among the different monomials, and the utility payoffs are obtained from the coefficients of the monomials times the number of monomials present in the polynomial function. A nonzero utility payoff is only obtained if the player selected the left action for every variable occurrence in the monomial that was drawn by the chance node.

## A.3 A NOTE ON OUR GUROBI IMPLEMENTATION

Our `Gurobi` method implements the constrained polynomial optimization problem (1). Gurobi supports nonlinear optimization if they can be implemented via quadratic constraints. Therefore, we

---

**Algorithm 5:** AMSGrad; AMS

1 Initialize learning rate $\eta > 0$, $\beta_1, \beta_2 \in [0,1)$, $\boldsymbol{x}^{(1)} \in \Delta(m)$, and $\boldsymbol{m}^{(0)}, \boldsymbol{v}^{(0)}, \hat{\boldsymbol{v}}^{(0)} = \boldsymbol{0}$

2 **procedure** GETX($\tilde{\boldsymbol{u}}^{(t)}$) **return** $\boldsymbol{x}^{(t)}$

3 **procedure** STEP($\boldsymbol{u}^{(t)}$)

4     $\boldsymbol{m}^{(t)} \leftarrow \beta_1 \boldsymbol{m}^{(t-1)} + (1 - \beta_1)\boldsymbol{u}^{(t)}$

5     $\boldsymbol{v}^{(t)} \leftarrow \beta_2 \boldsymbol{v}^{(t-1)} + (1 - \beta_2)(\boldsymbol{u}^{(t)})^2$

6     $\hat{\boldsymbol{v}}^{(t)} \leftarrow \max\{\hat{\boldsymbol{v}}^{(t-1)}, \boldsymbol{v}^{(t)}\}$

7     $\boldsymbol{x}^{(t+1)} \leftarrow \Pi_{\Delta(m), \sqrt{\hat{\boldsymbol{v}}^{(t)}}}\big(\boldsymbol{x}^{(t)} + \eta \boldsymbol{m}^{(t)}/\hat{\boldsymbol{v}}^{(t)}\big)$

---

follow the common practice to implement a product of variables $\Pi_{i=1}^k x_i$ in the objective function—which can arise from the terms $\mathbb{P}(z \mid \boldsymbol{x})$—as follows. We introduce $k - 1$ auxiliary variables $y_2, \ldots, y_k$, replace $\Pi_{i=1}^k x_i$ in the objective with the single variable $y_k$, and add the following $k - 1$ quadratic equations as additional constraints to the program: $y_2 = x_2 \cdot x_1$, and for each $i = 3, \ldots, k$: $y_i = x_i \cdot y_{i-1}$. If we work on a decision tree—as we do in this paper—we can reuse auxiliary variables efficiently in the following way. The same auxiliary variable $y_i$ will appear for all paths in the tree from the root node $h_0$ to some node $h$ in which the player had to play the actions associated $x_1, x_2, \ldots, x_i$ in that exact order to reach $h$. Multiple paths could fit to this description since the chance actions are not relevant here. Indeed, the value of $y_i$ in this optimization program will represent the player's contribution to the reach probability of such a node $h$ from root $h_0$ if the player plays according to $\boldsymbol{x}$.

### A.4 AMSGRAD

Reddi et al. (2018) proposed AMSGrad (AMS) as a fix to ADAM (Kingma & Ba, 2015), which may not converge in some stochastic convex optimization problems. AMS is described in Algorithm 5. The max operator, square root $\sqrt{}$, and division $/$ of vectors are to be interpreted element-wise. The projection operator $\Pi_{\Delta(m), \boldsymbol{v}} : \mathbb{R}^m \to \Delta(m)$ for a vector $\boldsymbol{v} \in \mathbb{R}^m_{\geq 0}$ is defined as

$$\boldsymbol{x} \mapsto \underset{\boldsymbol{y} \in \Delta(m)}{\operatorname{argmin}} ||\boldsymbol{y} - \boldsymbol{x}||_{\boldsymbol{v}} := \underset{\boldsymbol{y} \in \Delta(m)}{\operatorname{argmin}} \sqrt{\langle \boldsymbol{y} - \boldsymbol{x}, \boldsymbol{v}^T(\boldsymbol{y} - \boldsymbol{x})\rangle}\,.$$

We implement this projection onto the simplex efficiently using the algorithm by Helgason et al. (1980).

### A.5 DEPLOYMENT PHASE OF SIMULATION PROBLEMS

Figure 6 displays the subgame $\Gamma'$ representing the deployment phase of the simulation problem we start to describe in Figure 2 (right).

### A.6 A PROOF OF PROPOSITION 4

We adopt the notation of Algorithm 1. For any $\boldsymbol{x}^* \in \mathcal{X}$, we have

$$\sum_{t=1}^T [U(\boldsymbol{x}^*) - U(\boldsymbol{x}^{(t)})] \leq \sum_{t=1}^T \langle \nabla U(\boldsymbol{x}^{(t)}), \boldsymbol{x}^* - \boldsymbol{x}^{(t)}\rangle$$

$$= \sum_{t=1}^T \sum_{i=1}^n \langle \boldsymbol{u}_i^{(t)}, \boldsymbol{x}_i^* - \boldsymbol{x}_i^{(t)}\rangle \leq O_T(\sqrt{T}).$$

where we use, in turn: concavity of $U$, the definition of the utility vectors given to every information set, and the fact that RM is a regret minimizer (Hart & Mas-Colell, 2000). Dividing both sides by $T$ and taking $\boldsymbol{x}^*$ to be a global maximizer of $U$, it follows that there is always some iteration $1 \leq t \leq T$ for which $U(\boldsymbol{x}^*) - U(\boldsymbol{x}^{(t)}) \leq O_T(1/\sqrt{T})$.

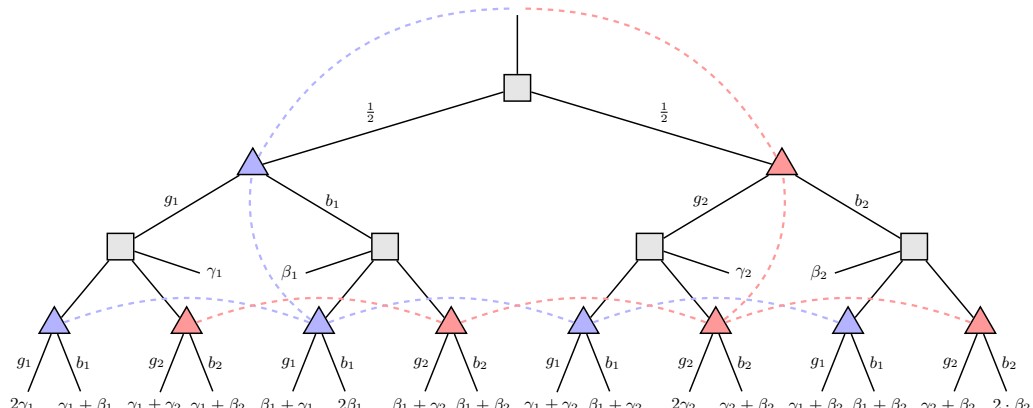

Figure 6: Deployment phase $\Gamma'$ of the more complex simulation problem with two scenarios given in Figure 2 (right). In deployment, the agent acts at least once and up to two times in total. The "good" and "bad" actions yield different immediate payoffs in different scenarios, and they contribute additively to the total payoffs at terminal nodes.

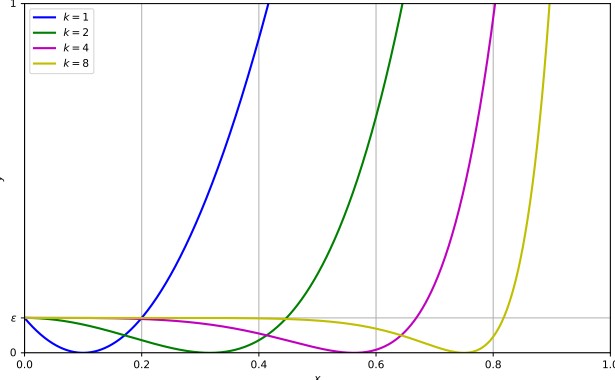

Figure 7: A family of polynomial optimization problems over the unit interval on which the `RM` and `GD` families of algorithms perform arbitrarily poorly.

### A.7 WHEN THE FIRST-ORDER OPTIMIZERS PERFORM POORLY

We find it quite surprising that the first-order optimizers we benchmark perform so well in terms of utility value in comparison to the global optimum found by `Gurobi`. Indeed, from a theoretical standpoint, we can give three examples in which the `RM` and/or `GD` families of algorithms converge to an arbitrarily bad value relative to the global optimum. To simplify explanations, we present all decision making under imperfect recall instances as maximization of a polynomial function over the 1-dimensional unit interval $[0, 1]$ instead. Theorem 1 describes how to efficiently construct the decision making under imperfect recall instance from that.

**Example 1: All converge to a bad value** Consider the $(\epsilon, k)$-parametrized function $f_{\epsilon,k}(x) = \frac{1}{\epsilon}(x^k - \epsilon)^2$ for $\epsilon > 0$ and $k \in \mathbb{N}$ over the unit interval $[0, 1]$. The function $f$ is plotted in Figure 7 for $\epsilon = 0.1$ and multiple values for $k$. In all cases, $f_{\epsilon,k}(x) \geq 0$, $f_{\epsilon,k}(0) = \epsilon$, and $f_{\epsilon,k}(1) > 1$ if we additionally restrict $\epsilon < \frac{3-\sqrt{5}}{2} \approx 0.382$. If an algorithm therefore converges to $x^* = 0$, we have found a family of instances for which the algorithm has achieved no more than MIN $+ \epsilon \cdot$ (MAX $-$ MIN) in value, where MAX and MIN represent the max and min values $f$ on $[0, 1]$ (or, respectively, the utility function on the 1-simplex). For our first-order methods, note that $f_{\epsilon,k}$ is strictly decreasing

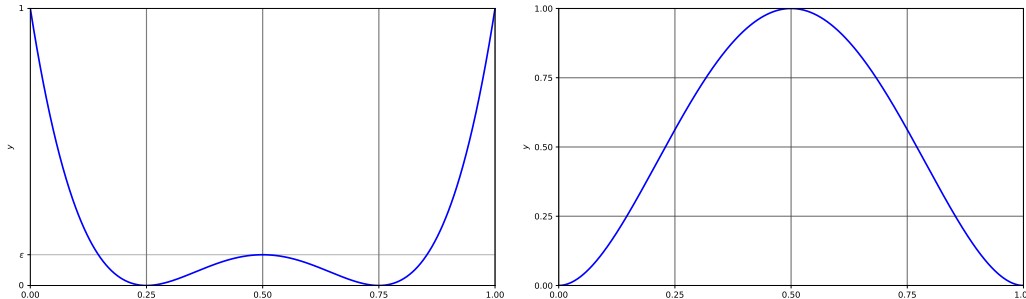

Figure 8: Examples in which particular first-order optimizers are likely to perform arbitrarily bad in value while others converge to the global optimum (if not initialized at $x^* = 1/2$ exactly). On the left, the gradient descent methods initialized closer to the middle will converge to the local optimum at $x^* = 1/2$ while the regret matching methods converge to the global optima. On the contrary, regret matching methods on the right will converge to the global minimum (which is also first-order maximal) while the gradient descent methods converge to the global optimum.

in the interval $J = [0, \epsilon^{1/k})$. If the RM and GD families of algorithms are therefore initialized to start in $J$, they will converge to $0$. Assuming we draw the initial point uniformly random from $[0, 1]$, this situation occurs with probability $\epsilon^{1/k}$. Therefore, we can first set the desired poor-performance parameter $\epsilon$, and then $k = k(\epsilon)$ to meet the desired probability confidence $\epsilon^{1/k}$, to obtain arbitrarily bad performance of the RM and GD families of algorithm with arbitrarily high probability in some instance.

**Example 2: gradient descent methods converge to a bad value**  Consider the function $f(x) = \left(\frac{1}{4} \cdot \frac{3}{4}\right)^2 \cdot (x - \frac{1}{4})^2 \cdot (x - \frac{3}{4})^2$ over the unit interval $[0, 1]$, as plotted in Figure 8 (left). Then GD, OGD, and AMS for any step size $\nu$ will converge to the local maximum $x^* = 1/2$ if initialized sufficiently close to it. We omit the extension of this singular example to a parametrized family of example—for which we would work with exponents of $f$ similar to Example 1 above—in order to increase the neighborhood of attraction around $x^* = 1/2$ and increase the difference $\frac{f(1/2)}{f(0)}$ of local optimum to global optimum. Independent of such extensions, the regret matching methods will always take such a large step at the first iteration that it reaches the global optimum at $x \in \{0, 1\}$ at the second iteration and stays there thereafter; that is, as long as the starting point is not exactly $x^* = 1/2$ or exactly one of the stationary points with value $0$ (for $f$, that is $\{\frac{1}{4}, \frac{3}{4}\}$).

**Example 3: Regret matching methods converge to a bad value**  We consider the fact that regret matching methods start off with very large steps in the direction of the gradients a blessing rather than a curse for the convergence speed of the first-order optimizer. However, we can also exploit that property when it comes to performance in terms of value achieved. Consider the function $f(x) = 16 \cdot x^2(1 - x)^2$ over the unit interval $[0, 1]$, as plotted in Figure 8 (right). Then as long as the regret matching methods do not start at the global optimum $x^* = 1/2$ exactly, they will grossly overshoot their first step in the direction of the gradient, and reach $x^{(2)} \in \{0, 1\}$ in the second iteration. Despite yielding the globally minimum value of $0$, the regret matching methods will be stuck there thereafter since those two points are also stationary points. The gradient descent methods for any step size $\nu$, on the other hand, will converge to the global optimum at $x^* = 1/2$ as long as they start sufficiently close it. Again, we omit the extension of this example to a parametrized class of examples that show how one can increase the neighborhood of attraction around $x^* = 1/2$.

A.8  ADDITIONAL EXPERIMENTAL DETAILS AND RESULTS

All experiments were run on a 64-core AMD Opteron 6272 processor. Each run was allocated one thread with a maximum of 16GBs of RAM. The commercial solver Gurobi requires a license to run on decision problems of nontrivial size. The result table of the experiments for the full set of benchmark decision problems is given in Table 3, which now also includes PRM experiments. We display "—" in the time column of Gurobi if it does not converge to the global optimum (up to a

| Problem | Gurobi value | Gurobi time | GD value | GD time | GD gap | OGD value | OGD time | OGD gap | AMS value | AMS time | AMS gap | RM value | RM time | RM gap | RM+ value | RM+ time | RM+ gap | PRM value | PRM time | PRM gap | PRM+ value | PRM+ time | PRM+ gap |
|---|---|---|---|---|---|---|---|---|---|---|---|---|---|---|---|---|---|---|---|---|---|---|---|
| Det-86 | 18.00 | 0.22s | 18.00 | 0.01s | — | 18.00 | 0.01s | — | 18.00 | 0.06s | — | 18.00 | 0.00s | — | 18.00 | 0.00s | — | 18.00 | 0.00s | — | 18.00 | 0.00s | — |
| Det-105 | 12.00 | 1.86s | 12.00 | 0.01s | — | 12.00 | 0.01s | — | 12.00 | 0.05s | — | 12.00 | 0.00s | — | 12.00 | 0.00s | — | 12.00 | 0.00s | — | 12.00 | 0.00s | — |
| Det-1k | 13.00 | 1m 24s | 13.00 | 0.13s | — | 13.00 | 0.07s | — | 13.00 | 1.04s | — | 13.00 | 0.32s | — | 13.00 | 0.36s | — | 13.00 | 0.38s | — | 13.00 | 0.41s | — |
| Det-1.8k | 22.00 | 2m 40s | 22.00 | 0.06s | — | 22.00 | 0.07s | — | 22.00 | 0.71s | — | 22.00 | 0.03s | — | 22.00 | 0.03s | — | 22.00 | 0.03s | — | 22.00 | 0.03s | — |
| Det-2.0k | 17.50 | 1m 42s | 17.50 | 0.03s | — | 17.50 | 0.05s | — | 17.50 | 0.20s | — | 17.50 | 0.03s | — | 17.50 | 0.03s | — | 17.50 | 0.03s | — | 17.50 | 0.03s | — |
| Det-8k | 16.67 | — | 16.62 | 13.05s | — | 16.62 | 2m 36s | — | 16.67 | 8m 1s | 3e-05 | 16.67 | 2m 39s | — | 16.67 | — | — | 16.67 | — | 0.001 | 16.67 | — | 0.007 |
| Det-10.6k | 12.84 | — | 12.70 | 24.87s | — | 12.70 | 5m 0s | — | 12.84 | 4m 44s | — | 12.84 | 6.41s | — | 12.84 | 6.74s | — | 12.84 | 15.53s | — | 12.84 | 14.48s | — |
| Det-10.7k | 20.20 | 16m 36s | 20.20 | 0.21s | — | 20.20 | 0.23s | — | 20.20 | 2.40s | — | 20.20 | 0.73s | — | 20.20 | 0.77s | — | 20.20 | 1.06s | — | 20.20 | 1.13s | — |
| Det-86k | 14.89 | — | 14.84 | — | 0.004 | 10.00 | — | 11.4 | 14.89 | — | 7e-05 | 14.89 | 2m 7s | — | 14.89 | 2m 4s | — | 14.89 | 7m 54s | — | 14.89 | 5m 50s | — |
| Det-130k | 15.53 | — | 15.37 | — | 8e-06 | 15.40 | 45m 59s | — | 15.53 | 57m 57s | — | 15.53 | 5m 38s | — | 15.53 | 2m 3s | — | 15.53 | — | 0.0001 | 15.53 | — | 8e-05 |
| Det-139k | 18.89 | — | 18.76 | 15m 24s | — | 18.76 | 16m 48s | — | 18.89 | 31m 26s | — | 18.89 | 1m 47s | — | 18.89 | 1m 50s | — | 18.89 | 3m 29s | — | 18.89 | 3m 10s | — |
| Det-718k | — | | 12.76 | — | 0.0005 | 12.69 | — | 0.005 | 12.84 | 2h 9m | — | 12.84 | 30m 55s | — | 12.84 | 31m 21s | — | 12.84 | — | 0.001 | 12.84 | — | 0.0008 |
| Det-1.002m | — | | 13.93 | — | 0.0003 | 13.90 | — | 0.005 | 13.96 | — | 1e-05 | 13.96 | 15m 36s | — | 13.96 | 17m 17s | — | 13.96 | 40m 10s | — | 13.96 | 35m 35s | — |
| Det-1.008m | — | | 12.64 | — | 0.0006 | 12.54 | — | 0.008 | 12.75 | — | 3e-05 | 12.75 | 33m 31s | — | 12.75 | 22m 38s | — | 12.75 | 54m 51s | — | 12.75 | 31m 28s | — |
| Det-2.1m | — | | 26.00 | — | 1e-05 | 25.96 | — | 0.02 | 26.15 | — | 0.002 | 26.15 | — | 0.003 | 26.15 | 3h 25m | — | 26.15 | — | 0.006 | 26.15 | — | 0.005 |
| Det-2.2m | — | | 16.20 | — | 0.002 | 15.93 | — | 0.02 | 16.36 | — | 0.0002 | 16.36 | 2h 22m | — | 16.36 | 3h 13m | — | 16.36 | — | 2e-06 | 16.36 | — | 5e-06 |
| Det-3.8m | — | | 15.66 | — | 0.003 | 15.14 | — | 0.03 | 15.80 | — | 0.0002 | 15.80 | — | 2e-06 | 15.80 | — | 5e-05 | 15.80 | — | 0.002 | 15.80 | — | 0.0003 |
| Det-4.0m | — | | 18.17 | — | 0.005 | 17.72 | — | 0.03 | 18.33 | — | 0.0005 | 18.34 | — | 2e-05 | 18.34 | 2h 55m | — | 18.34 | — | 0.005 | 18.34 | — | 0.005 |
| Det-4.1m | — | | 17.88 | — | 0.003 | 17.47 | — | 0.03 | 18.05 | — | 4e-05 | 18.06 | — | 2e-05 | 18.06 | — | 2e-05 | 18.06 | — | 0.003 | 18.06 | — | 0.0007 |
| Det-4.2m | — | | 19.98 | — | 0.003 | 20.07 | — | 0.003 | 20.15 | — | 0.0004 | 20.15 | — | 0.0004 | 20.15 | — | 2e-05 | 20.15 | — | 0.01 | 20.15 | — | 0.02 |
| Det-9m | — | | 23.16 | — | 0.004 | 22.71 | — | 0.02 | 23.45 | — | 0.004 | 23.45 | — | 0.0001 | 23.45 | — | 0.0001 | 23.45 | — | 0.0003 | 23.45 | — | 0.0004 |
| Det-10m | — | | 24.64 | — | 0.002 | 24.61 | — | 0.003 | 24.76 | — | 0.009 | 24.76 | — | 0.002 | 24.76 | — | 0.01 | 24.76 | — | 0.01 | 24.76 | — | 0.0008 |
| Det-18m | — | | 26.38 | — | 0.006 | 25.81 | — | 0.05 | 26.71 | — | 0.004 | 26.71 | — | 0.004 | 26.71 | — | 0.001 | 26.71 | — | 0.04 | 26.71 | — | 0.04 |
| Rand-7k | 0.53 | 25m 18s | 0.49 | 4.88s | — | 0.49 | 5.14s | — | 0.50 | 2.69s | — | 0.50 | 0.38s | — | 0.50 | 0.27s | — | 0.50 | 0.26s | — | 0.50 | 0.34s | — |
| Rand-11.9k | 1.00 | 1h 16m | 0.97 | 0.93s | — | 0.97 | 0.90s | — | 0.99 | 1.38s | — | 0.95 | 0.26s | — | 0.95 | 0.29s | — | 0.95 | 0.19s | — | 0.95 | 0.23s | — |
| Rand-12.2k | 1.00 | 1h 52m | 0.93 | 3.33s | — | 0.92 | 2.73s | — | 0.92 | 2.35s | — | 0.93 | 0.41s | — | 0.94 | 0.36s | — | 0.94 | 0.41s | — | 0.94 | 1.68s | — |
| Rand-24k | 0.72 | — | 0.66 | 7m 0s | — | 0.66 | 7m 46s | — | 0.66 | 4m 4s | — | 0.66 | 26.55s | — | 0.66 | 1m 3s | — | 0.66 | 1m 54s | — | 0.66 | 5m 5s | — |
| Rand-35k | 1.00 | — | 0.95 | 3.85s | — | 0.95 | 3.76s | — | 0.95 | 3.90s | — | 0.92 | 0.99s | — | 0.92 | 1.18s | — | 0.92 | 0.92s | — | 0.94 | 1.68s | — |
| Rand-42k | 0.69 | — | 0.55 | — | 0.01 | 0.55 | — | 0.01 | 0.64 | — | 0.0006 | 0.65 | — | 2e-06 | 0.65 | 5m 56s | — | 0.65 | — | 5e-06 | 0.65 | 3m 19s | — |
| Rand-165k | 0.37 | — | 0.96 | 19.77s | — | 0.97 | 18.48s | — | 0.99 | 28.90s | — | 0.96 | 4.33s | — | 0.97 | 4.95s | — | 0.96 | 5.24s | — | 0.90 | 4.02s | — |
| Rand-179k | 0.38 | — | 0.88 | — | 0.0003 | 0.88 | — | 1e-06 | 0.96 | 1m 7s | — | 0.94 | 5.97s | — | 0.93 | 10.27s | — | 0.93 | 6.66s | — | 0.91 | 7.31s | — |
| Rand-198k | 0.40 | — | 0.96 | 25.37s | — | 0.95 | 22.61s | — | 0.97 | 39.73s | — | 0.96 | 8.10s | — | 0.96 | 7.41s | — | 0.95 | 5.22s | — | 0.96 | 6.31s | — |
| Rand-1.2m | — | | 0.93 | 2m 46s | — | 0.93 | 2m 28s | — | 0.97 | 2m 17s | — | 0.96 | 35.86s | — | 0.97 | 36.07s | — | 0.96 | 31.74s | — | 0.96 | 31.09s | — |
| Rand-1.3m | — | | 0.96 | 4m 0s | — | 0.96 | 3m 17s | — | 1.00 | 8m 33s | — | 0.96 | 2m 26s | — | 0.98 | 54.77s | — | 0.98 | 1m 33s | — | 0.93 | 38.99s | — |
| Rand-2m | — | | 0.92 | 3m 53s | — | 0.93 | 3m 44s | — | 0.97 | 2m 1s | — | 0.94 | 59.29s | — | 0.93 | 1m 21s | — | 0.96 | 2m 1s | — | 0.96 | 58.84s | — |
| Rand-4m | — | | 0.94 | 13m 1s | — | 0.94 | 15m 39s | — | 0.96 | 29m 2s | — | 0.93 | 3m 12s | — | 0.92 | 4m 26s | — | 0.92 | 8m 41s | — | 0.93 | 4m 16s | — |
| Rand-6m | — | | 0.97 | 17m 34s | — | 0.97 | 15m 40s | — | 0.99 | 14m 23s | — | 0.98 | 2m 30s | — | 0.98 | 2m 9s | — | 0.98 | 2m 50s | — | 0.98 | 2m 10s | — |
| Rand-7m | — | | 0.97 | 22m 27s | — | 0.98 | 25m 7s | — | 0.99 | 11m 45s | — | 0.94 | 2m 13s | — | 0.93 | 2m 52s | — | 0.96 | 2m 55s | — | 0.97 | 3m 47s | — |
| Rand-13m | — | | 0.59 | — | 0.003 | 0.58 | — | 0.003 | 0.65 | 1h 40m | — | 0.63 | 19m 11s | — | 0.64 | 17m 31s | — | 0.64 | 20m 39s | — | 0.65 | 36m 42s | — |
| Rand-18m | — | | 0.97 | 2h 33m | — | 0.97 | 3h 0m | — | 0.99 | 1h 29m | — | 0.95 | 29m 45s | — | 0.97 | 24m 0s | — | 0.96 | 13m 8s | — | 0.97 | 14m 31s | — |
| Rand-23m | — | | 0.94 | 3h 37m | — | 0.93 | — | 0.0007 | 0.98 | 23m 10s | — | 0.98 | 23m 10s | — | 0.98 | 23m 5s | — | 0.98 | 16m 48s | — | 0.95 | 18m 2s | — |
| Sim-245 | 4.41 | 0.18s | 4.41 | 0.00s | — | 4.41 | 0.00s | — | 4.41 | 0.01s | — | 4.41 | 0.00s | — | 4.41 | 0.00s | — | 4.41 | 0.00s | — | 4.41 | 0.00s | — |
| Sim-438 | 7.21 | 0.41s | 7.21 | 0.00s | — | 7.21 | 0.00s | — | 7.21 | 0.01s | — | 7.21 | 0.00s | — | 7.21 | 0.00s | — | 7.21 | 0.00s | — | 7.21 | 0.00s | — |
| Sim-759 | 3.89 | 2.97s | 3.89 | 0.01s | — | 3.89 | 0.01s | — | 3.89 | 0.02s | — | 3.89 | 0.01s | — | 3.89 | 0.01s | — | 3.89 | 0.01s | — | 3.89 | 0.01s | — |
| Sim-3k | 6.25 | 1m 1s | 6.25 | 0.32s | — | 6.25 | 1.03s | — | 6.25 | 5.54s | — | 6.25 | 0.26s | — | 6.25 | 0.28s | — | 6.25 | 0.52s | — | 6.25 | 0.48s | — |
| Sim-7k | 8.58 | 1m 36s | 8.58 | 0.05s | — | 8.58 | 0.05s | — | 8.58 | 0.12s | — | 8.58 | 0.05s | — | 8.58 | 0.05s | — | 8.58 | 0.05s | — | 8.58 | 0.05s | — |
| Sim-13k | 10.38 | 4m 21s | 10.38 | 0.69s | — | 10.38 | 8.54s | — | 10.38 | 14.37s | — | 10.38 | 1.03s | — | 10.38 | 1.01s | — | 10.38 | 4.75s | — | 10.38 | 3.97s | — |
| Sim-34k | 10.44 | 1h 42m | 10.44 | 4.89s | — | 10.44 | 6.74s | — | 10.44 | 1m 9s | — | 10.44 | 2.52s | — | 10.44 | 2.81s | — | 10.44 | 5.03s | — | 10.44 | 5.01s | — |
| Sim-66k | 6.94 | 1h 31m | 6.94 | 5.63s | — | 6.94 | 8.70s | — | 6.94 | 1m 9s | — | 6.94 | 5.51s | — | 6.94 | 3.94s | — | 6.94 | 17.32s | — | 6.94 | 15.01s | — |
| Sim-105k | 4.40 | — | 4.40 | 18.60s | — | 4.40 | 1m 0s | — | 4.40 | 2m 35s | — | 4.40 | 2m 41s | — | 4.40 | 55.90s | — | 4.40 | 15m 18s | — | 4.40 | 10m 51s | — |
| Sim-125k | 14.47 | — | 14.48 | 12.70s | — | 14.48 | 19.76s | — | 14.48 | 4m 46s | — | 14.48 | 11.70s | — | 14.48 | 12.15s | — | 14.48 | 18.83s | — | 14.48 | 19.68s | — |
| Sim-226k | 8.57 | — | 9.70 | 2.16s | — | 9.70 | 4.28s | — | 9.70 | 9.81s | — | 9.70 | 1.52s | — | 9.70 | 1.50s | — | 9.70 | 1.49s | — | 9.70 | 1.48s | — |
| Sim-415k | 6.30 | — | 8.81 | 3.85s | — | 8.81 | 3.43s | — | 8.81 | 10.39s | — | 8.81 | 2.65s | — | 8.81 | 2.65s | — | 8.81 | 2.65s | — | 8.81 | 2.64s | — |
| Sim-441k | 11.79 | — | 13.57 | 57.23s | — | 13.57 | 1m 15s | — | 13.57 | 14m 15s | — | 13.57 | 36.88s | — | 13.57 | 33.94s | — | 13.57 | 2m 38s | — | 13.57 | 1m 35s | — |
| Sim-540k | 6.41 | — | 8.54 | 47.54s | — | 8.54 | 2m 37s | — | 8.54 | 15m 31s | — | 8.54 | 19.39s | — | 8.54 | 19.44s | — | 8.54 | 3m 48s | — | 8.54 | 3m 3s | — |
| Sim-866k | 8.77 | — | 10.49 | 2m 4s | — | 10.49 | 2m 27s | — | 10.49 | 17m 38s | — | 10.49 | 1m 31s | — | 10.49 | 1m 0s | — | 10.49 | 2h 34m | — | 10.49 | 2h 0m | — |
| Sim-1m | 4.14 | — | 4.77 | 5m 33s | — | 4.77 | 7m 2s | — | 4.77 | 29m 22s | — | 4.77 | 2m 14s | — | 4.77 | 2m 34s | — | 4.77 | 4m 16s | — | 4.77 | 4m 20s | — |
| Sim-1.7m | 11.05 | — | 13.33 | 10m 26s | — | 13.33 | 11m 16s | — | 13.33 | 3h 52m | — | 13.33 | 4m 3s | — | 13.33 | 4m 53s | — | 13.33 | 7m 22s | — | 13.33 | 7m 12s | — |
| Sim-1.9m | — | | 13.45 | 18.31s | — | 13.45 | 17.96s | — | 13.45 | 1m 2s | — | 13.45 | 12.36s | — | 13.45 | 12.19s | — | 13.45 | 12.48s | — | 13.45 | 12.47s | — |
| Sim-2.3m | — | | 11.09 | 22.01s | — | 11.09 | 21.88s | — | 11.09 | 1m 10s | — | 11.09 | 14.97s | — | 11.09 | 15.00s | — | 11.09 | 15.01s | — | 11.09 | 15.13s | — |
| Sim-4m | — | | 14.01 | 45m 5s | — | 14.01 | 41m 0s | — | 13.98 | — | 0.02 | 14.01 | 11m 36s | — | 14.01 | 7m 3s | — | 14.01 | 26m 45s | — | 14.01 | 21m 17s | — |

Table 3: Experimental results for the full set of benchmarks and the full set of algorithms. The winners per game in terms of value and convergence are highlighted in bold.

tolerance of $10^{-6}$ within the time limit), and "—" in its value column if it cannot even produce a "best-so-far" strategy within the time limit.[4]

The supplementary code contains the files that can generate decision problems with imperfect recall, solve them with the algorithms we discuss in Section 3, and plot their optimization progress. The particular benchmark instances of Table 3, together with experiments and plots regarding them, are available in the following Google drive link: https://drive.google.com/file/d/1v4WhJjRiZkOKegTvPeTXgBZtYLN_N1S7/view?usp=sharing.

In the writing of the code base, we have occasionally utilized LLM-based systems for code completion of simple or repetitive tasks. In those occasions, we ensure that we only include code snippets from the LLM that is correct to our understanding.

---

[4]This happens whenever `Gurobi` spends all of its time on *presolving*, and because we do not supply `Gurobi` with a strategy initialization.

