# OpenReview forum: "Decision Making under Imperfect Recall: Algorithms and Benchmarks"
_ICLR.cc/2026/Conference — Submitted to ICLR 2026_

### Official Review · Reviewer_iLnX · 2025-10-15

**Soundness:** 2
**Presentation:** 3
**Contribution:** 2
**Rating:** 4
**Confidence:** 4

**Summary:**

This paper makes two primary contributions. First, it introduces an extensive benchmark suite for decision-making under imperfect recall, with problem categories motivated by real-world concerns. Second, it provides a comprehensive empirical evaluation of algorithms for finding equilibria in these problems. The key finding is that the family of RM algorithms, particularly RM⁺, consistently and significantly outperforms standard first-order optimizers like PGD and its variants in terms of convergence speed, often by orders of magnitude. This establishes RM as a highly effective, parameter-free approach for this important class of problems.

**Strengths:**

1. The creation of the first large-scale benchmark for imperfect-recall decision problems is a significant and timely contribution. The problems are well-motivated and will serve as a valuable resource for the community, filling a critical gap.

2. The evaluation across 61 tasks and multiple algorithms is thorough, and the performance difference is so pronounced that it provides a clear, actionable takeaway for practitioners in this domain.

3. The paper is well-written and structured. It effectively introduces the necessary background, the benchmark design, and the algorithms, making it accessible to a broad audience.

**Weaknesses:**

1. The most significant weakness is the lack of a theoretical explanation for RM's superior performance. As noted, RM's efficacy in perfect-recall games is well-documented; however, its robust performance in the more complex and non-convex setting of imperfect-recall optimization demands deeper investigation. The paper would be greatly strengthened by providing theoretical intuition or analysis that explains why the RM update rule is so effective here, moving beyond the empirical demonstration [1-3].

2. The benchmark generation methods, while excellent for ensuring variety and avoiding cherry-picking, are largely random and syntactic. They may not fully capture the strategic, domain-specific abstractions [4, 5] or automated abstractions [6-8]. This could limit the immediate applicability of the findings to some real-world problems where imperfect recall is intentionally designed for efficiency.

3. Some novelty has been claimed. RM performs well in imperfect-recall game has been stated in [9], which asserted "CFR applied to any abstraction belonging to our general class results in a regret bound not just for the abstract game, but for the full game as well". Imperfect-recall abstraction performance has been tested in poker and randomly generated games with support [10, 11].

4. The paper does not propose a new algorithm but rather performs a systematic comparison of existing ones. While the empirical finding is highly valuable, the novelty is centered on the benchmarking and evaluation, not on methodological innovation.

If these issues can be resolved, I would be willing to raise my rating.

[1] Christian Kroer, Tuomas Sandholm. Imperfect-Recall Abstractions with Bounds in Games. EC 2016

[2] Christian Kroer, Tuomas Sandholm. A Unified Framework for Extensive-Form Game Abstraction with Bounds. NeurIPS 2018

[3] Nicolas S. Lambert, Adrian Marple, Yoav Shoham. On equilibria in games with imperfect recall. Games and Economic Behavior 2019

[4] Sam Ganzfried, Tuomas Sandholm. Potential-Aware Imperfect-Recall Abstraction with Earth Mover’s Distance in Imperfect-Information Games. AAAI 2014

[5] Noam Brown, Sam Ganzfried, Tuomas Sandholm. Hierarchical Abstraction, Distributed Equilibrium Computation, and Post-Processing, with Application to a Champion No-Limit Texas Hold'em Agent. AAMAS 2015

[6] Jirí Cermák, Viliam Lisý, Branislav Bosanský. Automated construction of bounded-loss imperfect-recall abstractions in extensive-form games. IJCAI 2020

[7] Boning Li, Zhixuan Fang, Longbo Huang. RL-CFR: Improving Action Abstraction for Imperfect Information Extensive-Form Games with Reinforcement Learning. ICML 2024

[8] Boning Li, Longbo Huang. Efficient Online Pruning and Abstraction for Imperfect Information Extensive-Form Games. ICLR 2025

[9] Marc Lanctot, Richard G. Gibson, Neil Burch, Michael Bowling. No-Regret Learning in Extensive-Form Games with Imperfect Recall. ICML 2012

[10] Jirí Cermák, Branislav Bosanský, Viliam Lisý. An Algorithm for Constructing and Solving Imperfect Recall Abstractions of Large Extensive-Form Games. IJCAI 2017

[11] Jirí Cermák, Branislav Bosanský, Michal Pechoucek. Combining Incremental Strategy Generation and Branch and Bound Search for Computing Maxmin Strategies in Imperfect Recall Games. AAMAS 2017

**Questions:**

1. The strong performance of RM in both perfect and imperfect-recall settings suggests a fundamental advantage. Can you provide any intuition or preliminary analysis for why the RM update rule is so well-suited to constrained optimization over the product-of-simplex domain, even in non-convex problems like these?

2. How do you envision the community building upon your benchmark? Do you believe your conclusions about RM would hold for more strategically-generated, domain-specific imperfect-recall abstractions, and would you encourage the creation of such instances within your framework?

3. Could you test different imperfect-recall algorithms in common games? Could you test how different degrees of forgetting affect the solution?

---

> ### Author Response · Authors · 2025-11-21
>
> We thank the reviewer for their time and effort dedicated to providing valuable feedback, questions, and comments on our paper. We are glad to hear that they appreciated the motivation of our benchmark and breadth of our experiments. Below, we will respond to their questions point by point, starting with a clarification.
>
> $\phantom{.}$
>
> ### Touching on the multiple comments that RM has already been shown to perform well in decision making under imperfect recall in the past, where the reviewer refers to the literature on imperfect-recall abstractions
>
> We will attempt to disambiguate these thoughts with three separate responses. In short, our work _is complementary_ to the advances on imperfect-recall game solving mentioned by the reviewer, which revolves around game abstractions to deal with bounded memory. Neither the techniques nor the motivation of that literature applies to our general setting of imperfect recall. We invite the reviewer to let us know if any questions remain.
>
> 1. Abstractions: Our paper discusses the related work on imperfect-recall abstractions, and what role RM has played in this line of work—including many of the references provided by the reviewer—in the introduction and in Appendix Section A. (We want to remark that reference [3] seems mistakenly cited in this specific context; it is an economics theory paper that is neither computational nor on RM methods.) All other references by the reviewer fall within the line of work on imperfect-recall game abstractions for the purpose of managing memory constraints (such as in AI poker). That literature has focused exclusively on well-behaved subclasses of imperfect-recall games, which, in particular, exclude absentmindedness ([11] states that directly; the restrictive class of (chance-relaxed) skew well-formed games from [1,9] exclude multiple phenomena of imperfect recall, including absentmindedness; [10] only merges perfect-recall infosets that are at the same depth; and [2] focuses on acyclic infoset structures). The sole motivation of that line of work is to find computationally tractable (imperfect-recall) abstractions of a _perfect-recall_ game, from which we can recover good strategies and performance downstream for the original perfect-recall game.
>
> 2. In contrast, we study solving decision problems with imperfect recall, and specifically absentmindedness, for its own sake. Our work is the first to do that from an empirical lens / at scale. The applications that our work is motivated by come from the area of trustworthy AI (testing and evaluation of AI agents via simulated environments, handling of sensitive or private data, etc.). The structure of these decision problems, as we argue, are themselves modeled best as exhibiting imperfect recall and absentmindedness, making them _inherently_ difficult to solve even at small scale (at least, as we show, to global optimality).
>
> In the setting of game abstractions, on the other hand, it is up to the abstraction method to decide in what convenient way the agent should forget particular information in order to ease some of the memory load. In particular, the methods developed in that line of work (and assumptions imposed on the structure of imperfect recall) do not apply to the problems that motivate our work.
>
> 3. Algorithmic Contribution: Our work is the first to apply regret matching methods to general imperfect-recall problems, and more broadly, nonlinear optimization problems in practice. This is in contrast to (two-player) zero-sum _perfect-recall_ games in which RM has been widely popular. Prior work has also explored RM within the CFR framework for the narrow subclasses of imperfect-recall problems described above. Unfortunately, Waugh et al. [2009, Section “Challenges of Imperfect Recall”] have shown fundamental obstacles to extending CFR to general imperfect-recall settings (Footnote 3, in Appendix Section A). In short, notions such as “in the past” and “playing actions in order to reach an infoset” (once, or multiple times?) are not well-defined anymore.
>
> - In light of this, our main algorithmic contribution lies in extending the scope of RM methods to any imperfect-recall problem (by applying RM to the _agent form_), and showing that RM and RM+ attain strong performance in nonlinear optimization. The fact that they perform so well in our benchmarks is surprising, since we are working with a class of problems that is fundamentally harder than the classes for which CFR is known to work (We recall Proposition 2 and that decision making under imperfect recall is expressive enough to fully capture global optimization of polynomial objective functions over simplex domains.)

---

> ### Author Response · Authors · 2025-11-21
>
> ### “The strong performance of RM in both perfect and imperfect-recall settings suggests a fundamental advantage. Can you provide any intuition or preliminary analysis for why the RM update rule is so well-suited to constrained optimization over the product-of-simplex domain, even in non-convex problems like these?”
>
> As the reviewer points out, regret matching is known to have good practical performance in two-player zero-sum perfect-recall games, and there is some conceptual understanding of why this is the case. For example, Chakrabarti et al. [2024] discuss a property of RM that is believed to be important to practical performance: namely, at the infoset level, RM is invariant to step sizes, and thus does not require careful step size tuning in the same way that gradient descent does. Their explanation applies verbatim to the imperfect recall setting. While this is not a full theoretical explanation, we will include it in the revised version of the paper to give some intuition for why RM might perform well empirically compared to GD variants, both in zero-sum games and beyond.
>
> That being said, we agree with the reviewer that proving theoretically why RM is a formidable first-order optimizer in nonlinear problems is an exciting avenue for future research, arguably, ignited by the practical insights from our paper. It turns out, the fundamental question of whether RM is _guaranteed_ to asymptotically converge to a first-order optimal point in the first place is open.
> 1. One observation we can make (and include in the paper) is that if the objective function (to be maximized) is concave, then RM is guaranteed to converge to a global maximum of that function because RM is a regret minimizer.
> 2. Narrowing down on the front of nonconcave objectives, even a special case—convergence of RM to Nash equilibria in identical-interest games—has been open for more than 20 years (at least, since Hart and Mas-Colell, 2003), so resolving that is challenging. Absentmindedness—which is the focus of our paper—makes the problem even harder by making variables in the utility function appear in higher polynomial degree.
>
> We believe that our paper provides strong motivation for a follow-up theoretical examination of the regret matching family of algorithms in nonlinear optimization.
>
> $\phantom{.}$
>
> ### “How do you envision the community building upon your benchmark? Do you believe your conclusions about RM would hold for more strategically-generated, domain-specific imperfect-recall abstractions, and would you encourage the creation of such instances within your framework?”
> We kept the format of our benchmark quite flexible so that future work can build on it; any decision problem or multiplayer game on a finite tree can be added to our benchmark using the tools we already provide, whether with _perfect recall_ or with _imperfect recall_. Our implemented algorithms work out-of-the-box for single-player problems, and are straightforwardly extendable to the multiplayer setting (with the template we present in Algorithm 1).
>
> As examples, we can discuss three interesting research directions that can be carried out using our benchmark suite and open-source code:
> 1. What optimization or learning algorithms should we use to solve _multiplayer_ games of imperfect recall. There are many open questions here about what solution concept we should aim for. Nash equilibria, for example, may not even exist anymore [Wichardt, 2008].
> 2. Advancing the state-of-the-art imperfect-recall game abstractions. For example, it would be interesting to study empirically when (exactly) a method can get away with abstracting a game to the point that it exhibits absentmindedness, without losing (much) value in the original perfect-recall game. The RM methods we study could perform quite well here, not only because they performed well in our benchmark, but also because RM within the CFR framework are known to perform well under narrow forms of imperfect recall [1,9].
> 3. Informing the design of trustworthy AI systems in practice. How should one optimally design testing and evaluation schemes in order to quantifiable reduce the possibility for misaligned agents to pass evaluations [Chen et al. 2024]? How much sensitive and private information can companies and institutions commit to forgetting before their ability to perform their tasks degrades significantly?
>
> $\phantom{.}$
>
> ### “Could you test different imperfect-recall algorithms in common games?”
> What common games does the reviewer have in mind here?

---

> ### Author Response · Authors · 2025-11-21
>
> $\phantom{.}$
>
> ### “Could you test how different degrees of forgetting affect the solution?”
> Higher degrees of absentmindedness increase the curvature of the landscape of the objective function, that is, it becomes more hilly. As such, it can become increasingly difficult for first-order optimizers to successfully find the global optimum.
>
> In the generated decision problems in our benchmark suite, we already experiment with varying the parameter that controls the degree of absentmindedness. With regards to the performance of first-order optimizers, we can only identify a trend for the problem class on detection with privacy constraints: here, RM methods perform comparable to the gradient descent variants for small degrees of absentmindedness, outperform them by a wide margin for large degrees, and outperform them even more strongly for medium degrees of absentmindedness. We would like to note here that increasing the parameter for the degree of absentmindedness—all other parameters held equal—also increases the size of the problem instance. That last remark holds for the class of simulation problems.
>
>
> $\phantom{.}$
>
> ### References
> [Chakrabarti et al. 2024] Darshan Chakrabarti, Julien Grand-Clément, and Christian Kroer. 2024. Extensive-form game solving via blackwell approachability on treeplexes. In Proceedings of the 38th International Conference on Neural Information Processing Systems (NeurIPS '24).

---

> > ### Comment · Reviewer_iLnX · 2025-11-23
> >
> > As the author states, most abstract methods can only operate within specific games or fixed levels, proving incapable of handling absent-minded scenarios across arbitrary games. Nevertheless, I maintain that the previous algorithms, especially automatic abstraction methods, can still perform absentminded abstraction for any game. Specifically, [6] can derive improved absentminded abstractions from any existing one, while [8] can perform absentminded abstraction on information sets sharing identical expected values in the current and several future steps. I still believe that many abstraction methods can be applied to absentmindedness.
> >
> > By the way I contend that absentminded abstraction is not arbitrary or random; it must align with real-world scenarios. For instance, an absentminded driver may possess only seconds of memory, allowing us to perform absentminded abstraction in SIMULATION PROBLEMS based on actions from the preceding few steps. An absentminded lost individual may only perceive scenes within their field of vision, enabling us to conduct absentminded abstraction in SUBGROUP DETECTION based on subgraph structures.
> >
> > Moreover, I concur that absentmindedness and imperfect-recall are not entirely synonymous, and this paper primarily focuses on the former. However, the current version (particularly the title and abstract) contains extensive references to imperfect-recall, which may readily give rise to misunderstanding. Given the close association between imperfect-recall and absentmindedness, the authors ought at the very least to mention that the RM algorithm demonstrates superior performance in imperfect-recall under specific conditions.

---

> > > ### Author Response · Authors · 2025-11-27
> > > **We want to make clear that our work is not about abstractions**
> > >
> > > We thank the reviewer for their follow-up feedback and engagement with our work. We want to make clear that our work, benchmark suite, and algorithms _are not about abstractions_.
> > >
> > > In the setting of our paper, there is _no_ underlying (perfect-recall) game for which imperfect-recall abstractions are being constructed. Instead, our goal is to solve a game that _inherently_ features imperfect recall. This is a completely distinct, and much harder, problem (for a given size of the decision tree). Indeed, Kroer and Sandholm [2016, second paragraph above Theorem 3.3] clearly separate it from the work on abstractions: in that literature (which includes the papers cited by the reviewer), the goal is to build a smaller game that is _easy to solve_. Thus, the forms of imperfect recall encountered in the abstraction literature have been relatively benign, and maintain the property that equilibria can be efficiently computable. While that literature is related to our paper in the sense that both deal with imperfect recall, the motivations, techniques, and evaluations involved in the abstraction literature are completely different from ours.
> > >
> > > Also, our benchmark suite introduces decision problems that can exhibit _all forms_ of imperfect recall, which is why we have considered our title and abstract appropriate. Our benchmark includes absentmindedness, but also other phenomena such as forgetting what action oneself or some other player (_e.g._ chance) has taken in the past, or whether one has been in another particular infoset in the past. Furthermore, we investigate algorithms that can solve a decision problem with any kind of imperfect recall.
> > >
> > > ### The methods from citations [6] and [8] cannot be applied to our benchmark suite
> > >
> > > Citations [6,8] introduce abstraction methods that may generate imperfect-recall abstractions. The methods themselves are not applicable to our benchmark suite because we are not concerned with solving a game through (imperfect-recall) abstractions. As a subroutine the methods of [6,8] use CFR to solve the imperfect-recall abstraction (and [8] also tests a fictitious play method). As we described before and in the paper, CFR is not applicable to decision problems that exhibit absentmindedness. And indeed, we can confirm that the imperfect-recall abstractions that appear in [6,8] are designed without absentmindedness.
> > > 1. The imperfect-recall abstractions for poker in [8] are produced by merging two infosets whose public states are the same (Section 4.3, second paragraph, condition 1). In particular, this excludes absentmindedness.
> > > 2. The two methods introduced by [6; better described in the full version, found under https://arxiv.org/abs/1803.05392 and published in Artificial Intelligence 2020] start with an initial coarse abstraction, and gradually refine the abstraction (by breaking off infosets) whenever solutions and regrets from the underlying perfect-recall decision problem do not translate well into the imperfect-recall abstraction. If we tried to apply it to absentminded problems such as the absentminded driver, then the algorithms would immediately break apart the absentmindedness. This renders the method unsuitable for us. Moreover, [6] only experiments with an initialized imperfect-recall abstraction that is obtained from merging infosets at the same depth, and therefore does not encounter any decision problems with absentmindedness throughout the paper (see Section 4.1.1, and line 4 of Section 5.1).

---

### Official Review · Reviewer_pAUu · 2025-10-30

**Soundness:** 3
**Presentation:** 3
**Contribution:** 3
**Rating:** 6
**Confidence:** 4

**Summary:**

The paper shows that variants of regret matching may be a suitable heuristic for solving imperfect recall decision problems. It argues why solving these problems is important, proposes several scalable benchmarks, and compares the achieved value and computations requirement of the several iterative algorithms to the optimal solution computed by Gurobi. They conclude that on a range of domains, RM based solutions perform close to optimum (where computable) in a fraction of time.

**Strengths:**

* The paper explains reasonably well why imperfect recall problems are important.
  * It presents a sufficiently wide range of experiments to convince me that RM based methods are worth considering for this problem.
  * It promises public release of a range of problems, that may become a common benchmark for future solvers
  * No need to tuning parameters compared to GD

**Weaknesses:**

* Even the local convergence is supported only empirically
  * Negative examples where local optima are bad are hidden in the appendix and referenced to a different paper instead of clearly discussing the limitations
  * The main results in Table 1 are hard to interpret


Further suggestions:

The paper at places seems to make claims about general constraint polynomial optimization, not just imperfect recall games (L99,L255, 278), which looks confusing to me. Either make clear what wider class of optimization problems you suggest RM should work in, or stick to imperfect recall decision making.

There are more advanced algorithms for solving imperfect recall games, such as [A]. It would be nice to comment on why those cannot be used, instead of "Gurobi". Also, for the "Gurobi" method, I would appreciate the exact formulation you used to solve the problem, in the appendix. This (and related) papers also introduce a parametrized class of imperfect recall random games, which can be related to yours.

[A] Čermák, Jiři, Branislav Bošanský, and Michal Pěchouček. "Combining incremental strategy generation and branch and bound search for computing maxmin strategies in imperfect recall games." Proceedings of the 16th Conference on Autonomous Agents and MultiAgent Systems. 2017.

I believe it would be better to clearly describe the examples of imperfect recall decision making problems, where RM fails in the main body of the text.

It is very hard for me to read Table 1. Can you add some aggregate statistics over the columns or something that would clearly compare the algorithms without staring at the table for 10 minutes?

**Questions:**

Why are Gurobi times for Rand-*k not reported? Why are OGD times not reported for some games, where the value is reported?

I did not fully get the domain motivation of the detection domain. Can you elaborate more on a specific real world situation modeled by the game?

---

> ### Author Response · Authors · 2025-11-21
>
> Thank you for the encouraging feedback on our paper. We appreciate the time and effort dedicated to providing insightful questions and valuable suggestions. We will respond to them in chronological order below:
>
> $\phantom{.}$
>
> ### ““Even the local convergence is supported only empirically”
> First, we want to note two remarks here:
> 1. Our negative example in Appendix Section A.4 shows that our first-order methods—including RM methods—may get trapped at poor local optima. In particular, it is an example that shows that they might not converge to a global optimum. This is true more generally for any efficient algorithm, as it is NP-hard to get anywhere close to the globally optimal value (Proposition 2, showing it for a constant additive error).
> 2. One observation we can include in the paper is that if the objective function (to be maximized) is concave, then RM converges to a global maximum of that function because RM is a regret minimizer.
>
> That being said, we agree with the reviewer that proving RM’s convergence to first-order optimal points in the nonconcave setting is an exciting avenue for future research, arguably, ignited by the practical insights from our paper. As it turns out, even a special case of this problem—convergence of RM to Nash equilibria in identical-interest games—has been open for more than 20 years (at least, since Hart and Mas-Colell, 2003), so resolving that is challenging. Absentmindedness—which is the focus of our paper—makes the problem even harder by making variables in the utility function appear in higher polynomial degree. We believe that our paper provides strong motivation for a follow-up theoretical examination of the regret matching family of algorithms in nonlinear optimization.
>
> $\phantom{.}$
>
> ### “Negative examples where local optima are bad are hidden in the appendix and referenced to a different paper instead of clearly discussing the limitations” and “I believe it would be better to clearly describe the examples of imperfect recall decision making problems, where RM fails in the main body of the text.”
> Thank you for this suggestion, we are happy to move the figure and discussion into the main body if the reviewers think it is a good idea. A quick clarification question: Is the reviewer’s comment pointing at our paper reference to Tewolde et al. [2023] in L861? If so, we want to note that Tewolde et al. do not give any negative examples; we cite them here for describing the equivalence of decision making under imperfect recall and polynomial optimization over simplices.
>
> In response to the reviewer’s comment, we want to highlight that we haven’t observed proper failure modes of RM and other first-order optimizers in practice, that is, with regards to the utility value they achieved in the experiments with our large and diverse benchmark suite (cf. our main takeaways in Section 5). That being said, Reviewer YU3n expressed interest in the failure modes as well, so we designed another negative example which might be of interest here.
>
> Namely, RM methods can get trapped at poor local optima while the gradient descent variants we consider converge to the global optimum. Take the polynomial function $f(x) := 16 \big( x (1-x) \big)^2$ which maps the interval [0,1] to [0,1] again. Its minima are at $x = 0$ and $x = 1$ with objective value 0 and gradient 0, and its only global = local maximum over [0,1] is at $x^{\star} = 1/2$ with objective value 1. Then for the projected gradient descent algorithm with fixed step size, there exist close enough initialization points $x$ to $x^{\star}$, such that PGD will converge to $x^{\star}$. Informally, if you start close to a (non-degenerate) local maximum, you ought to converge to it. RM methods do not have this property: no matter how close to $x^{\star}$ you initialize, as long as it is not exactly $x^{\star}$, the first step will overshoot and land on the boundary, where you will remain stuck because the boundary points are first-order optimal.
> What is going on in that example is that RM methods make a first gradient step of “infinite” size, and they will keep step sizes generally large at the first few iterations. Generally, we consider this _a blessing_ rather than a curse for the convergence speed of RM. (Indeed, in that last example, RM methods converged to a first-order optimal point within one iteration!) By the design of our construction, however, that particular first-order point is arbitrarily bad in objective value. In a similar way, we can construct examples where the gradient descent variants are very likely to get trapped at a poor local optimum while the RM methods all converge consistently to the global optimum.

---

> ### Author Response · Authors · 2025-11-21
>
> ### “The main results in Table 1 are hard to interpret” and “It is very hard for me to read Table 1. Can you add some aggregate statistics over the columns [...]?”
> We welcome this feedback for improving our presentation. It is not quite clear what aggregate statistics work well here, since the different classes of decision problems have different possible utility ranges (which are not known a priori) and different computational complexities in terms of convergence speed. Averaging convergence speeds is further complicated since many algorithms do not reach full convergence in large instances, in which case we report the CDT gap.
>
> $\phantom{.}$
>
> ### “The paper at places seems to make claims about general constraint polynomial optimization, not just imperfect recall games (L99,L255, 278), which looks confusing to me. Either make clear what wider class of optimization problems you suggest RM should work in, or stick to imperfect recall decision making.”
> This is a great point to clarify. We use decision making under imperfect recall interchangeably with polynomial optimization over simplices because they are computationally equivalent in a formal sense! In particular, our development of RM for decision making under imperfect recall extends analogously to nonlinear optimization over products of simplices generally, which is why we presented the RM methods as general first-order optimizers for nonlinear objective functions (possibly beyond polynomials) over simplex domains.
>
> Formally, optimizing utility (resp. finding CDT equilibria) in decision making under imperfect recall reduces to optimizing a polynomial function (resp. finding a first-order optimal point = KKT point of it) over a product of simplices. Prior work has also established a reduction in the other direction [Gimbert et al. 2020, Tewolde et al. 2023], from which they derived NP-hardness and inapproximability results for finding optimal strategies (Proposition 2), as well as CLS-completeness results for finding CDT equilibria. We remark that occurrences of variables in higher polynomial degree correspond to the presence of absentmindedness in the corresponding decision problem.
>
> $\phantom{.}$
>
> ### “There are more advanced algorithms for solving imperfect recall games, such as [Cermak et al. 2017]. It would be nice to comment on why those cannot be used, instead of "Gurobi". [...] This (and related) papers also introduce a parametrized class of imperfect recall random games, which can be related to yours.”
> Thank you for the reference. In short, these algorithms cannot be applied to our decision problems because that line of work assumes narrow forms of imperfect recall.
> More concretely, that paper excludes absentmindedness in order for their algorithm to work, whereas our benchmark focuses on absentmindedness. Cermak et al. exclude absentmindedness because their Double Oracle-style algorithm only searches for pure best responses. In the presence of absentmindedness, however, one might very well need to randomize (even in single-player settings) in order to find global or local optima, as illustrated in the popular and simple absentminded driver example [Piccione and Rubinstein 1997]. The underlying reason is that absentmindedness introduces nonlinearities of the form of variables appearing with higher polynomial degree in the utility function.
>
> More generally, past papers on practical solving of imperfect-recall games have exclusively (1) focused on game abstractions of very large perfect-recall games (such as poker), (2) assumed narrow forms of imperfect recall that exclude absentmindedness, and (3) evaluated the solutions based on whether they can be lifted to good strategies in the original, perfect-recall game. Therefore, the decision problems that were studied in that line of work are not of primary interest to our benchmark suite. In contrast, the motivation for our benchmark suite and evaluations are problems that _intrinsically_ exhibit imperfect recall—and absentmindedness specifically—coming from problems around trustworthy AI (testing and evaluation of AI agents via simulated environments, handling of sensitive or private data, etc.).
>
> $\phantom{.}$
>
> ### “Also, for the "Gurobi" method, I would appreciate the exact formulation you used to solve the problem, in the appendix.”
> Gurobi can handle a polynomial objective function over a product of simplices, which—as we describe in Section 2.1—is the problem we are facing. Would the reviewer want us to state that polynomial optimization problem somewhere in the paper in a mathematical environment, or to see the implementation detail of how to put a higher-degree polynomial objective function into Gurobi?
> 1. For the former, we can include Equation (11) of Tewolde et al. [2023] in our revised version
> 2. To explain the latter, we can represent an example such as $x^2y$ with two auxiliary variables $z, z’$, objective $z$, and additional quadratic constraints $z = z’ \cdot y$ and $z’ = x \cdot x$.

---

> ### Author Response · Authors · 2025-11-21
>
> ### “Why are Gurobi times for Rand-*k not reported? Why are OGD times not reported for some games, where the value is reported?”
> In line with the procedure we describe in Section 5, we only report times when convergence has been reached within the time limit (for Gurobi, “convergence” means “termination”). Even the small Rand-*k instances turned out to be too large for Gurobi to find and prove global optimality within 4 hours. For first-order optimizers (including OGD), we can always report a utility value, namely the value of the latest iterate. In contrast, Gurobi can take so much time on _presolving_  that it never produces a feasible point within the time limit to produce a utility value (cf. Footnote 4). Lastly, we cannot report times for first-order optimizers that do not reach convergence within the time limit. In those cases, we therefore report the KKT Gap = CDT Gap at the last iteration instead, which forms the natural indicator for how close the algorithms have gotten to the convergence condition.
>
> $\phantom{.}$
>
> ### “I did not fully get the domain motivation of the detection domain. Can you elaborate more on a specific real world situation modeled by the game?
> Thanks for asking. Continuing from our example in the introduction: Suppose we have a medical AI system tasked with identifying suitable candidates for blood donation. Potential candidates would be reluctant to share confidential information about their health status—HIV status, medical history, etc.—unless the AI has been designed to delete any knowledge regarding patients that were deemed unsuitable, thus exhibiting imperfect recall. Moreover, the AI system is provided with a meaningful connection graph which captures some notion of relationships or similarity between the potential candidates (possibly based on known family or social ties, physical characteristics, geography, etc.): “if person A is suitable, then person B, who is connected to A, is also more likely to be suitable, because suitable candidates come in groups”. The AI system now has to calculate what sequence of potential candidates to test for suitability (1) given its time or cost budget, and, more importantly, (2) given its knowledge of the candidates it successfully identified as suitable in the past, without remembering who else turned out to be unsuitable. This flavour of problem—identifying subjects with particular characteristics, being constrained by dealing with sensitive data—can also be found in countless other real-world domains. For example, we can construct similar examples to the one above
> - in marketing, where a firm wants access to a database of contact addresses of potential customers, and the potential customers want to keep their contact addresses private/unknown from the firms to which they have expressed no interest,
> - in cybersecurity, where an institution could want to examine how wide-spread a particular cybervirus is in a computer network, and owners of the computers are reluctant to give access to their computers unless their personal data is forgotten afterwards, or
> - in the economics of innovation (described in the introduction), where investment firms would like to understand the ideas and implementation of the invented technology, but the inventor does not want the underlying details of its product to be known to anyone who hasn’t yet entered into a beneficial investment agreement with the inventor.
>
> In all of these examples, we are envisioning that the firm/institution that has to deal with the sensitive or private data has automated their decision processes by delegating (some) decision making to an AI agent. That AI agent can then be designed to verifiably have imperfect recall, and furthermore, be absentminded.

---

> ### Comment · Reviewer_pAUu · 2025-11-25
>
> Thank you for the example. I believe that it is important. However, it further deepens the confusion about whether you are solving imperfect information games or polynomial optimization. You should clarify it the introduction of the paper. Also, consider adding a more detailed explanation of the content of Tewolde et al to preliminaries.  I did not go deeper into that paper. I just assumed they present a construction that demonstrates that GD/RM will not converge, if they show it can represent polynomial optimization. And yes, having the full optimization problem including the constraints you use for the "Gurobi solution" somewhere in a referenceable form would IMO be useful.
>
> I understand that coming up with aggregate statistics for Table 1 is not easy. Still, I believe you should do it. You may use the bounds on the value and cut-off times. I am sure you will come up with something that would be better than nothing at all.
>
> I understand you are focusing on absentmindedness, which prevents you from comparing to many other algorithms and I am fine with that. However, it could be states little more clearly.
>
> I believe the paper needs substantial edits to explain the relationship, the detection domain, integrate the negative examples, so I am waiting with further discussion until we have a revision.

---

### Official Review · Reviewer_YU3n · 2025-10-31

**Soundness:** 3
**Presentation:** 3
**Contribution:** 3
**Rating:** 6
**Confidence:** 4

**Summary:**

This paper introduces the first benchmark suite for decision-making under imperfect recall, a setting in which an agent may forget previously acquired information. The authors design three parametric classes of problems—simulation games (for AI safety and testing), subgroup detection under privacy constraints, and random decision problems—implemented using the LiteEFG framework. They evaluate a range of first-order optimization algorithms for computing Causal Decision Theory (CDT) equilibria, including projected gradient descent (PGD), AMSGrad, and a newly introduced family of regret matching (RM) algorithms for nonlinear constrained optimization. Experiments across 61 instances show that RM-based algorithms, especially RM+, consistently outperform gradient-based optimizers in convergence speed (often by orders of magnitude) while achieving comparable or better objective values. The paper provides both theoretical motivation and large-scale empirical results, establishing RM algorithms as strong general-purpose optimizers beyond their traditional game-theoretic use.

**Strengths:**

S1. Novel benchmark suite fills a clear gap in the literature.

S2. Bridges two research communities —game theory and optimization—by adapting regret matching to nonlinear constraints.

S3. Extensive experiments across diverse problem types and scales, with clear reporting of performance metrics and runtime.

S4. Strong practical insight: RM+ shows remarkable stability and speed, potentially influencing solver design for large imperfect-information systems.

S5. Clarity and reproducibility: methodology, code availability, and hyperparameter details are well-documented.

**Weaknesses:**

**W1.** Lack of theoretical guarantees for RM convergence in general nonconvex constrained settings.  While empirical evidence is compelling, a formal proof (even partial or asymptotic) would strengthen the claim that RM can act as a “first-order optimizer.”

---

**W2.** Limited analysis of failure cases.  The appendix briefly mentions instances in which RM converges to poor local optima, but a deeper investigation of when and why this occurs would enhance understanding.

---

**W3.** Connection to learning-based methods (e.g., policy gradient or actor-critic under imperfect recall) is only briefly mentioned; integrating these discussions could expand relevance to the broader ICLR audience.

---

**W4.** Benchmark diversity: all benchmarks are tabular; extending to function approximation or continuous domains is left for future work.

**Questions:**

**Q1.** How sensitive are RM variants to initialization or stochastic noise compared to gradient-based methods?

**Q2.** Could the authors theoretically link RM’s updates to mirror descent or other known first-order schemes under certain convexity assumptions?

**Q3.** In the simulation benchmark, does varying the simulation–deployment ratio qualitatively change RM’s convergence behavior?

**Q4.** Would integrating RM with adaptive step sizes (like AMSGrad) further improve performance?

**Q5.** Could the benchmark be extended to multi-agent imperfect-recall games (beyond single-player decision problems)?

---

> ### Author Response · Authors · 2025-11-21
>
> We thank the reviewer for their encouraging feedback on our paper. We appreciate the time and effort dedicated to providing insightful questions and valuable suggestions. We will respond to them in chronological order below:
>
> $\phantom{.}$
>
> ### “[...] While empirical evidence is compelling, a formal proof (even partial or asymptotic) would strengthen the claim that RM can act as a “first-order optimizer.””
>
> One observation we can include in the paper is that if the objective function (to be maximized) is concave, then RM converges to a global maximum of that function because RM is a regret minimizer.
>
> Nonetheless, we agree with the reviewer that the nonconcave setting is an exciting avenue for future research, arguably, ignited by the practical insights from our paper. As it turns out, even a special case of this problem—convergence of RM to Nash equilibria in identical-interest games—has been open for more than 20 years (at least, since Hart and Mas-Colell, 2003), so resolving that is challenging. Absentmindedness—which is the focus of our paper—makes the problem even harder by making variables in the utility function appear in higher polynomial degree. We believe that our paper provides strong motivation for a follow-up theoretical examination of the regret matching family of algorithms in nonlinear optimization.
>
> $\phantom{.}$
>
> ### "Limited analysis of failure cases. The appendix briefly mentions instances in which RM converges to poor local optima, but a deeper investigation of when and why this occurs would enhance understanding.”
> Before touching on failure modes, we want to highlight the strong performance of RM and other first-order optimizers with regards to the utility value they achieved in the experiments with our large and diverse benchmark suite:
> 1. In all cases in Table 1 where Gurobi has converged to the globally optimal utility value, the first order methods reach that optimal value as well.
> 2. In larger scale problems where we do not know the optimal value, we have observed that “RM+ and AMSGrad oftentimes attain higher values than GD and OGD, and almost never less.”
>
> While we haven’t observed failure modes in practice, we can still construct negative examples.
> 1. As the reviewer notes, we give a concrete example in the appendix to show that the first-order methods we test—including RM methods—may get trapped at poor local optima. This is true more generally for any efficient algorithm, as it is NP-hard to get anywhere close to the globally optimal value (Proposition 2, showing it for a constant additive error).
> 2. We can also give another illustrative example in which RM methods get trapped at poor local optima while our gradient descent methods converge to the global optimum. Take the polynomial function $f(x) := 16 \big( x (1-x) \big)^2$ which maps the interval [0,1] to [0,1] again. Its minima are at $x = 0$ and $x = 1$ with objective value 0 and gradient 0, and its only global = local maximum over [0,1] is at $x^{\star} = 1/2$ with objective value 1. Then, given a projected gradient descent algorithm with fixed step size, there exist close enough initialization points $x$ to $x^{\star}$, such that PGD will converge to $x^{\star}$. Informally, if you start close to a (non-degenerate) local maximum, you ought to converge to it. RM methods do not have this property: no matter how close to $x^{\star}$ you initialize, as long as it is not exactly $x^{\star}$, the first step will overshoot and land on the boundary, where you will remain stuck because the boundary points are first-order optimal.
>
> What is going on in that example is that RM methods make a first gradient step of “infinite” size, and they will keep step sizes generally large at the first few iterations. Generally, we consider this _a blessing_ rather than a curse for the convergence speed of RM. (Indeed, in that last example, RM methods converged to a first-order optimal point within one iteration!) By the design of our construction, however, that particular first-order point is arbitrarily bad in objective value. In a similar way, we can construct examples where the gradient descent variants are very likely to get trapped at a poor local optimum while the RM methods all converge consistently to the global optimum.
>
> We are happy to discuss these examples with figures in the revised version of the paper since we are not aware of such examples in the literature.

---

> > ### Author Response · Authors · 2025-11-21
> >
> > ### “Connection to learning-based methods (e.g., policy gradient or actor-critic under imperfect recall) is only briefly mentioned” and “extending to function approximation or continuous domains is left for future work.”
> > As the reviewer recognizes, we think that these are potentially interesting technical extensions to our work. That being said, the prominent problems studied in decision making and games with absentmindedness so far have been naturally discrete in nature, which forms the basis of our benchmark suite and the scope of the motivation of this paper. We did not run into the need for function approximation or gradient estimation methods in our experiments since utilities and gradients continued to be efficiently computable in instances with 10+ million nodes.
> >
> > $\phantom{.}$
> >
> > ### “How sensitive are RM variants to initialization or stochastic noise compared to gradient-based methods?”
> > With respect to stochastic initializations, we observe that for small instances, the gradient descent methods tend to have a higher variance in the convergence process, whereas in large-scale instances that are not from the simulation type, the RM variants (together with AMSGrad) tend to have a higher variance. Since we can compute exact gradients, there is no stochastic noise in our first-order methods.
> >
> > $\phantom{.}$
> >
> > ### “Could the authors theoretically link RM’s updates to mirror descent or other known first-order schemes under certain convexity assumptions?”
> > The paper by Farina et al. [2021] makes a connection between regret matching and mirror descent. The upshot is that RM+ can be obtained by running mirror descent on a suitable “lifted space” related to Blackwell approachability. Yet, it’s unclear whether that connection can be used to analyze the convergence of RM+ because the transformation that maps the lifted strategies to actual strategies is not well-behaved.
> >
> > $\phantom{.}$
> >
> > ### “In the simulation benchmark, does varying the simulation–deployment ratio qualitatively change RM’s convergence behavior?”
> > This is a great question. In the benchmark instances we released, we already vary that ratio while keeping all other parameters fixed. Yet we do not observe any clear trends in these experiments. We would like to note here that the payoffs are determined stochastically, and that while changing that simulation–deployment ratio, one inevitably also changes the size of the problem instance.
> >
> > $\phantom{.}$
> >
> > ### “Would integrating RM with adaptive step sizes (like AMSGrad) further improve performance?”
> > The RM methods are _parameter-free_, which are part of their appeal. They don’t have any step sizes that one has to tune carefully, and yet RM+ showed dominant performance in our experiments.
> >
> > $\phantom{.}$
> >
> > ### “Could the benchmark be extended to multi-agent imperfect-recall games (beyond single-player decision problems)?”
> > Yes, we kept the format of our benchmark flexible enough so that any decision problem or multiplayer game on a finite tree can be added to our benchmark using the tools we already provide, whether with _perfect recall_ or with _imperfect recall_. Our implemented algorithms are also straightforwardly extendable to the multiplayer setting (with the template we present in Algorithm 1). We believe multi-agent imperfect-recall games are an interesting avenue for future research. However, there are many open questions in this setting around what solution concept one might aim for. Nash equilibria may not even exist anymore [Wichardt, 2008], and, to our knowledge, there haven’t been any solution concepts proposed for general games with imperfect recall/absentmindedness that would be amenable to efficient learning algorithms (e.g., involving correlation).
> >
> > $\phantom{.}$
> >
> > ### References
> > [Farina et al., 2021] Gabriele Farina, Christian Kroer, and Tuomas Sandholm. Faster Game Solving via Predictive Blackwell Approachability: Connecting Regret Matching and Mirror Descent. Proceedings of the AAAI Conference on Artificial Intelligence 35 (6):5363-71, 2021.
> >
> > [Wichardt, 2008] Philipp C. Wichardt. Existence of Nash Equilibria in Finite Extensive Form Games with Imperfect Recall: A Counterexample. Games Econ. Behav., 63(1):366–369, 2008.

---

### Official Review · Reviewer_rBEf · 2025-11-01

**Soundness:** 3
**Presentation:** 2
**Contribution:** 2
**Rating:** 4
**Confidence:** 4

**Summary:**

The paper aims to study algorithms for imperfect recall decision making problems.  They introduce three parameterized benchmark environments and propose to assess algorithms by how well they compute CDT equilibria, which amount to finding first order optimal, the very thing projected gradient descent algorithms do.  They then explore the use of regret matching for nonlinear constrained optimization and find that it outperforms projected gradient descent in this class of problems.

**Strengths:**

The paper is tackling an interesting problem.  The empirical evaluation is quite large, spanning not only multiple, but also quite different environments.

The observations on the performance of RM algorithms in these problems is interesting, and deserves further attention and investigation.

**Weaknesses:**

It is not entirely clear to me what is the contribution of the paper. It seems to be spread across presenting benchmarks, arguing for a particular evaluation metric, and then a set of evaluations that suggest RM methods should get more attention.  I'm not sure it does the first two that effectively.  I'm not sure the latter is sufficiently impactful, given the long history of imperfect recall being explored in games (using exactly RM-based algorithms; see work coming out of the AI poker competitions from 2007 to 2017).

Some of the motivation for imperfect recall "will play a key role in AI" are unconvincing.  Solutions to imperfect recall seek to cope with the forgotten information, but may effectively do so by using other remembered cues to recover that information.  Many of the examples aim to force forgetting as a principle (for privacy or simulation testing adherence purposes), where an agent's recovery of that information would be harmful to the mechanism's purpose itself.  I do think imperfect recall is an important thing to study purely from the framing of bounded rationality and long-lived agents.  Remembering one's entire past is clearly impractical beyond very short horizon settings.

Given that the paper purports to introduce benchmarks for evaluation, I would expect more discussion of their choice of evaluation metric.  They seem to settle on CDT equilibria without much justification beyond the fact that optimality (or approximation of it) is NP-complete.  That may be, but then that doesn't make just any tractable evaluation metric a good choice.  In fact, maybe it suggests the endeavor is hopeless, and we would be better off finding tractable subclasses (see work on well-formed games; Lanctot et al., ICML 2012; Kroer et al., EC 2016).  Why would reaching a CDT equilibria be desirable or sufficient?

Algorithm 1 is incomplete.  What is GetX?  What is Step?

"for generic nonconvex optimization problems such as ours"... Huh?  How is your subclass of non-convex optimizations somehow generic?!

I don't understand the subgroup detection problem at all.  What's the sequence of decisions?  What's the utilities?

"The confidence intervals represent the 30th and 70th percentile"  Does that mean these represent 40% confidence intervals?!?  I'm not sure you have enough iterations to be expecting to produce accurate confidence intervals, but would use is a 40% confidence interval anyway?  It seems like you're just trying to show your plots have tight shaded regions!  With under 15 samples, showing dots for all the samples in addition to the mean, or showing bars giving min and max would be more fair and honest.

Another related concept to CDT equilibria is "action deviations" from the literature on regret algorithms (Morrill, ICML 2021).  It may be worth relating to that work.

**Questions:**

You mentioned the Gurobi solver is guaranteed to find the global optimum if it terminates, yet it failed to do so in a couple of your problems. How can that be the case?  There's also times it terminated but you didn't report a running time.

---

> ### Author Response · Authors · 2025-11-21
>
> We thank the reviewer for their time and effort dedicated to providing valuable feedback, questions, and comments on our paper. We are glad to hear that they appreciated the breadth of our benchmark and experiments. Below, we will respond to their questions and comments; starting from a broader perspective, and continuing with more specific clarifications.
>
> $\phantom{.}$
> ### On the impact of our work, relative to other works for solving imperfect-recall games
>
> Our work _is complementary_ to the advances mentioned by the reviewer on imperfect-recall game solving which followed the AI poker competitions and revolves around game abstractions to deal with bounded memory (cf. the opening paragraph in our introduction, and Appendix Section A).
> 1. That line of work has focused exclusively on mapping _perfect-recall_ games (e.g. poker) to well-behaved (and computationally easier to handle) subclasses of imperfect-recall games, with the sole motivation to recover good strategies and performance downstream for the original perfect-recall game.
>
> 2. In contrast, we study solving decision problems with imperfect recall, and specifically absentmindedness, for its own sake. Our work is the first to do that from an empirical lens / at scale. The applications that our work is motivated by come from the area of trustworthy AI (testing and evaluation of AI agents via simulated environments, handling of sensitive or private data, etc.). The structure of these decision problems, as we argue, are themselves modeled best as exhibiting imperfect recall and absentmindedness, making them _inherently_ difficult to solve even at small scale (at least, as we show, to global optimality).
>
> In the setting of game abstractions, on the other hand, it is up to the abstraction method to decide in what convenient way the agent should forget particular information in order to ease some of the memory load. In particular, the methods developed in that line of work (and assumptions imposed on the structure of imperfect recall) do not apply to the problems that motivate our work.
>
> $\phantom{.}$
> ### Clarifying the contributions of our paper
>
> Our work has two core contributions:
>
> 1. Benchmark Suite: Decision making under imperfect recall, and specifically absentmindedness, have been extensively studied since the early years of game theory (cf. Kuhn, 1953, and other citations discussed in the appendix). So far, this has been done with pen and paper. Our work is the first to develop an empirical framework for decision making under imperfect recall through a flexible suite of benchmarks and implemented algorithms. As we pointed out above, earlier empirical work on imperfect-recall abstraction focused only on narrow subclasses, while our scope here is much broader. With it, we contribute to a flourishing line of work focusing on benchmarks for large-scale solving of strategic decision problems and games, complementing the work that has been done _with perfect recall_ [Nudelman et al. 2004, Liu et al. 2013, Weiss et al. 2017, Samvelyan et al. 2019, Carroll et al. 2019, Lanctot et al. 2020, Bard et al. 2020, Papoudakis et al. 2021].
> 2. Algorithmic Contribution: Our work is the first to apply regret matching methods to general imperfect-recall problems, and more broadly, nonlinear optimization problems in practice. This is in contrast to (two-player) zero-sum _perfect-recall_ games in which RM has been widely popular. Prior work has also explored RM within the CFR framework for a narrow class of imperfect-recall problems. Unfortunately, Waugh et al. [2009] have shown fundamental obstacles to extending CFR to general imperfect-recall settings. In light of this, our main algorithmic contribution lies in extending the scope of RM methods to any imperfect-recall problem (by applying RM to the _agent form_), and showing that RM and RM+ attain strong performance in nonlinear optimization. The fact that they perform so well in our benchmarks is surprising, since we are working with a class of problems that is fundamentally harder than two-player zero-sum games. (We recall Proposition 2 and that decision making under imperfect recall is expressive enough to fully capture global optimization of polynomial objective functions over simplex domains.)

---

> > ### Author Response · Authors · 2025-11-21
> >
> > ### Why convergence to a CDT equilibrium is an appropriate evaluation metric
> >
> > In short, there are two key reasons we work with CDT equilibria:
> > 1. This concept was introduced and extensively motivated in numerous prior papers in economics, computer science, and philosophy (we refer to Appendix Section A), and
> > 2. The CDT equilibrium corresponds exactly to first-order optimal points (KKT points = stationary points of gradient descent), and as such, our evaluation metric for first-order methods is consistent with much of the contemporary work in nonlinear optimization and machine learning.
> >
> > In other words, CDT equilibria are the most natural solution concept from an optimization standpoint.
> >
> > While optimal strategies are a natural solution concept in decision problems with imperfect recall, they are NP-hard to compute (Proposition 2). More importantly, the experiments with our benchmark show that optimal strategies are also hard to find in practice: Gurobi, a state-of-the-art commercial solver, failed to compute optimal strategies beyond very small instances.
> >
> > $\phantom{.}$
> >
> > ### “Some of the motivation for imperfect recall ‘will play a key role in AI’ are unconvincing.”
> > We want to push back on this point. As appreciated by Reviewers pAUu and iLnX, imperfect-recall problems are well-motivated in applications concerning handling sensitive information, simulating and evaluating AI agents, and building trustworthy AI. These are some of the reasons why there has been a flourishing line of work that focuses on decision problems with imperfect recall, and absentmindedness specifically.
> >
> > $\phantom{.}$
> > ### “... How is your subclass of non-convex optimizations [that is, the class of decision problems under imperfect recall] somehow generic?!”
> > Great question. It has been shown that any optimization problem of a polynomial function over a product of simplices (possibly nonconvex) can be captured by utility maximization in a decision problem under imperfect recall [Gimbert et al. 2020, Tewolde et al. 2023]. This is how prior work has derived NP-hardness and inapproximability results for finding optimal strategies (Proposition 2). We remark that occurrences of variables in higher polynomial degree correspond to the presence of absentmindedness in the corresponding decision problem.
> >
> > $\phantom{.}$
> > ### “... the Gurobi solver is guaranteed to find the global optimum if it terminates, yet it failed to do so in a couple of your problems. How can that be the case? There's also times it terminated but you didn't report a running time.”
> > Indeed, Gurobi is guaranteed to terminate in finite time, and once it has terminated, it is guaranteed to have found a globally optimal solution. So when we state “Gurobi fails to converge beyond small instances” we mean that Gurobi fails to reach termination in a reasonable amount of time, hence, we cannot report a running time. In that case, we can still report the best utility value that Gurobi has found so far whenever it managed to complete its presolving stage within the time limits (Footnote 4).
> > In any case, the slow performance with a state-of-the-art global optimizer such as Gurobi calls for more scalable approaches, motivating our study of gradient descent and regret matching methods.
> >
> > $\phantom{.}$
> > ### Related work Morrill et al. [2021]
> > We thank the reviewer for bringing up the paper by Morrill et al. [2021]. Morrill et al. study notions of regret in extensive-form games based on action deviations, but restrict their attention to games _with perfect recall_.  In particular, questions such as “if I deviate to another action now, what effect does it have on the times in the past and future when I might enter that same infoset again”—to which causal decision theory (CDT) aims to provide an answer—do not arise under perfect recall (nor under the well-behaved subclasses of imperfect-recall games discussed far above). Moreover, we do not use regret matching as a _regret minimizer_ in our work, but we apply it to the incomparable problem of finding first-order optimal points (=strategies); and discover that regret matching performs exceptionally well for that task.

---

> > > ### Author Response · Authors · 2025-11-21
> > >
> > > ### Clarifying the sequence of decisions in the subgroup detection problem
> > > If the reviewers think this is a good idea, we can include another figure in the appendix for visualizing how the extensive-form decision problem might concretely look like in an example of subgroup detection on a graph (Figure 3). Starting from the example in Figure 3 (left), and paraphrasing Section 4.2, the decision problem works as follows. The agent starts with one action per node of the graph, representing the act of _selecting the node_. A parameter in our benchmark defines how many rounds the agent can select nodes. At the start, random chance selects small subsets of nodes (unbeknownst to the agent) to form desirable _subgroups_, and the agent aims to maximize the number of selected nodes belonging to a subgroup. More precisely, the agent receives an additional utility of $\lambda_i$ for each node of subgroup $i$ it has selected. If the agent selects a node that is part of a subgroup, the agent will observe and remember for the rest of the decision problem that it has already picked that node before, effectively removing that action henceforth from the available ones. If it selects a node that is not part of a subgroup, it will forget that fact in future rounds. In particular, that node remains available to be selected again.
> > >
> > > $\phantom{.}$
> > >
> > > ### On Algorithm 1
> > > Algorithm 1 is a _template_ for a first-order optimizer; defining the GetX and Step methods equates to implementing a concrete optimizer. Algorithms 2 and 5 describe this for Projected Gradient Descent, OGD, and AMSGrad, and Algorithms 3 and 4 demonstrate how we can fit regret matching methods into this framework.
> > >
> > > $\phantom{.}$
> > >
> > > ### Confidence intervals in Figure 4
> > > We will change the shaded region to represent min and max values as per the reviewer’s suggestion. This did not affect our takeaways from the experiments. (We refer to the following screenshot until the revised version of the paper is ready: https://drive.google.com/file/d/1NbvS4nZZAmDeBHK1ve7w1DN6KngrWelw/view?usp=sharing)
> > > The other suggested alternative—visualizing all samples as individual dots—would overload the figure, since we test 4 RM methods and 3 gradient descent methods (each with 4 different hyperparameter configurations) on 12 seeds.
> > >
> > >
> > > $\phantom{.}$
> > >
> > > All in all, we thank the reviewer again for this great discussion.
> > >
> > > $\phantom{.}$
> > >
> > > ### References
> > > [Bard et al. 2020] Nolan Bard, Jakob N Foerster, Sarath Chandar, Neil Burch, Marc Lanctot, H Francis Song, Emilio Parisotto, Vincent Dumoulin, Subhodeep Moitra, Edward Hughes, et al. The hanabi challenge: A new frontier for ai research. Artificial Intelligence. 2020.
> > >
> > > [Carroll et al. 2019] Micah Carroll, Rohin Shah, Mark K. Ho, Thomas L. Griffiths, Sanjit A. Seshia, Pieter Abbeel, and Anca Dragan. On the utility of learning about humans for human-AI coordination. NeurIPS (Neural Information Processing Systems). 2019.
> > >
> > > [Gimbert et al., 2020] Hugo Gimbert, Soumyajit Paul, and B. Srivathsan. A bridge between polynomial optimization and games with imperfect recall. In Proceedings of the 19th International Conference on Autonomous Agents and MultiAgent Systems, AAMAS ’20, page 456–464, Richland, SC, 2020. International Foundation for Autonomous Agents and Multiagent Systems.
> > >
> > > [Lanctot et al. 2020] Marc Lanctot, Edward Lockhart, Jean-Baptiste Lespiau, Vinicius Zambaldi, Satyaki Upadhyay, Julien Pérolat, Sriram Srinivasan, Finbarr Timbers, Karl Tuyls, Shayegan Omidshafiei, et al. Openspiel: A framework for reinforcement learning in games. arXiv:1908.09453. 2019.
> > >
> > > [Liu et al. 2013] Miao Liu, Xuejun Liao, and Lawrence Carin. Online expectation maximization for reinforcement learning in POMDPs. IJCAI (International Joint Conference on Artificial Intelligence). 2013.
> > >
> > > [Papoudakis et al. 2021] Georgios Papoudakis, Filippos Christianos, Lukas Schäfer, and Stefano V. Albrecht. Benchmarking multi-agent deep reinforcement learning algorithms in cooperative tasks. NeurIPS (Neural Information Processing Systems) Track on Datasets and Benchmarks. 2021.
> > >
> > > [Samvelyan et al. 2019] Mikayel Samvelyan, Tabish Rashid, Christian Schroeder de Witt, Gregory Farquhar, Nantas Nardelli, Tim GJ Rudner, Chia-Man Hung, Philip HS Torr, Jakob Foerster, and Shimon Whiteson. The starcraft multi-agent challenge. AAMAS (International Joint Conference on Autonomous Agents and Multiagent Systems). 2019.
> > >
> > > [Weiss et al. 2017] Michael Weiss, Benjamin Lubin, and Sven Seuken. SATS: A Universal Spectrum Auction Test Suite. AAMAS (International Joint Conference on Autonomous Agents and Multiagent Systems). 2017.

---

> > > ### Comment · Reviewer_rBEf · 2025-11-26
> > > **Remain Unconvinced**
> > >
> > > After reading the author response, my assessment is largely unchanged.
> > >
> > > First, I appreciate the explanation for the Gurobi performance numbers.
> > >
> > > Second, I'll offer a few comments in response.
> > >
> > > * I think the reference to the explorations of imperfect recall in poker is being misunderstood.  The actual abstraction choices being used in competitive poker programs did not fall under the well-formed classes cited above.  Those works were largely driven by the success of RM algorithms to handle pretty arbitrarily constructed imperfect recall abstractions, where there was no theory.  While concepts like skew-well-formed extended the theory, the abstractions being used were not skew-well-formed, and yet similar to your observations RM within these settings was still highly performant.  It is true that typical poker abstractions were not "absent-minded", so there is somewhat of a distinction in what can be concluded.
> > >
> > > * The arguments for the exploration of imperfect recall (i.e., privacy, trustworthiness, etc.) do not mesh well with the arguments supporting CDT as an appropriate solution concept.  If our goal is to develop advances appropriate for the described applications, than the justifications for a solution concept need to justify that it suffices to address the application.  But the paper, appendix (and as far as I could tell the cited papers), and rebuttal keep arguing for CDT's appropriateness on computability/theoretical grounds.
> > >
> > > * The reason I brought up Morrill's work is that action deviations seem to me to be giving a CDT-like guarantee (as it treats every decision made as an independent regret minimizer, making no perfect-recall requirements to minimize.)

---

> > > > ### Author Response · Authors · 2025-11-28
> > > >
> > > > We thank the reviewer for the clarifying comments on their initial review.
> > > >
> > > >
> > > > ### Our experimental findings for RM methods are novel and important even though there has been a flourishing line of research on abstractions
> > > >
> > > >
> > > > We agree with the reviewer that RM methods (within the framework of CFR) have been successfully used in imperfect-recall abstraction settings not covered by theory. Nonetheless, we believe that the experimental finding—that RM methods perform well in the general setting of imperfect-recall problems—is novel and important because our setting differs starkly from the abstraction literature in terms of motivation and design.
> > > >
> > > > The goal of abstractions is to build a smaller game that is easy to solve. Thus, the forms of imperfect recall encountered in the abstraction literature have been relatively benign, and maintain the property that equilibria can be efficiently computable. Our goal, instead, is to solve a game that inherently features imperfect recall, which is completely distinct, and much harder.
> > > >
> > > > In particular, we experiment with arbitrary forms of imperfect recall and have a special focus on problems that can also exhibit absentmindedness. To the best of our knowledge, this work is the first to include such problems.
> > > >
> > > >
> > > > Moreover, the literature on poker and abstractions has focused on RM methods as _regret minimizers_ for imperfect-recall abstractions. Our work finds that they are formidable first-order optimizers in nonlinear problems. These two are incomparable. (first-order optimal points can exhibit positive regret versus the global optimum, and no-regret sequences can be far away from first-order optimal points.)
> > > >
> > > >
> > > > ### Why the CDT equilibrium is an appropriate solution concept
> > > >
> > > >
> > > > We want to clarify that our perspective is reflected more accurately by the following line of thinking: The only goal of the experiments in our paper is to find methods that can achieve high utility in decision problems with imperfect recall. For that purpose, we have collected a few prominent algorithm contestors by going through the optimization and learning literature, and adapted them to our application domain. We have seen in our experiments that Gurobi indeed achieves the highest utility if it reached termination, and that RM+ and AMSGrad achieve the highest utilities most consistently in the full benchmark suite. Since many of our algorithms are first-order optimizers that get stuck at first-order optimal points, we can _also_ investigate how long they take to reach convergence. This characteristic indicates how little computational resources it requires to run an algorithm and subsequently check its performance cap. RM+ generally performs best in regards to convergence speed, _and_ it is a top contestor for achieving the highest utility value.
> > > >
> > > >
> > > >
> > > > ### Morrill et al.’s work remains only distantly related
> > > >
> > > >
> > > > Morrill et al.’s concept of an action deviation is related to the CDT equilibrium concept to the extent that they both work in the _agent-form_ of the decision tree, that is, both view each infoset as controlled by a separate player. We are happy to add a note to that in the revised paper. As mentioned earlier, there remain two important differences in that Morrill et al. (a) restrict themselves to the perfect-recall setting (page 3 second paragraph) and (b) consider notions of correlated equilibria, which is why they focus on regret minimization. The CDT equilibrium, on the other hand, uses as a starting point the Nash equilibrium concept in the agent-form of a decision problem, and relaxes it _in the presence of_ absentmindedness to allow for first-order optimality instead (cf. Tewolde et al., 2023). (As described earlier, this is different from the relaxation of a correlated equilibrium; in fact, we are not aware of work that has developed a notion of correlated equilibrium for decision problems and games with absentmindedness.)

---

### Author Response · Authors · 2025-12-03
**A Revised Version of the Paper and a Final General Response**

We thank the reviewers for their service and helpful feedback.

Unfortunately, our ongoing discussions with reviewers rBEf and iLnX were cut short by the Nov 27 incident at a point in time where we could clarify any misunderstandings about the motivation of our paper, and where were very much looking forward to their updated feedback. Nevertheless, **we have uploaded a new revision that addresses issues raised by each reviewer**, and highlighted key revisions in blue text color. In the appendix—where most of the changes took place—only the title of the added paragraphs/subsections were highlighted in blue.

**Revision highlights include:**
1. **More context on the theoretical work that our paper builds on, as well as a more extensive related literature paragraph on game abstractions. These will hopefully foster a better understanding for why our experimental contributions are novel and suggest wide-ranging implications to constrained nonlinear optimization.**
2. **Two illustrative examples plus experimental insights that give explanations for when exactly regret matching methods perform strongly / poorly in terms of value relative to the gradient descent methods.**

Below we briefly explain the key additions. The individual points and questions of each reviewer are addressed as separate responses.
1. We separate our work clearly from the literature on game abstractions, which has developed solvers that leverage imperfect-recall abstractions. (Two sentences in the contribution section and Footnote 1, with pointers to a related work paragraph in Appendix Section A.1.) In short, imperfect-recall abstractions are, by design, very structured, and intend to accurately model an underlying perfect-recall game. **Our work**, on the other hand, **focuses on solving decision problems that inherently feature imperfect recall, and studies methods that can tackle _all forms of imperfect recall_, including absentmindedness**. We also expanded our discussion on why (regret matching) methods introduced by the game abstractions literature _cannot be applied_ to our setting and problem instances, and that our work is—to the best of our knowledge—the first to experimentally consider regret matching methods in nonlinear optimization problems and as as a first-order optimizer. To be accurate, our paper falls into a line of work on imperfect recall that is separate from and has long preceded the literature on game abstractions [Kuhn,1953; Piccione & Rubinstein, 1997; Lambert et al., 2019; Berker et al., 2025; Kovarik et al., 2025a].
2. Reviewers rBEf and pAUu would have liked to see more explicitly why decision making under imperfect recall is capturing polynomial optimization over a product of simplices in its full generality. To that end, we have added a paragraph and a formal theorem (Theorem 1, together with references to the original papers)  to the Preliminaries Section 2.1, as well as a proof sketch and visualization to Appendix Section A.2.
3. We present and discuss **two additional illuminating examples** in Appendix Section A.7 in the revised version, **which highlight the exact extreme cases where gradient methods perform arbitrarily bad while the regret matching methods converge to the global optimum, and vice versa**. We do so because Reviewers YU3n and pAUu appreciated the illustrative example we gave in the original submission, and asked for more of such insights.
4. In response to Reviewer iLnX’s question, we have also added a paragraph to Section 5 that gives some conceptual explanation to the effectiveness of regret matching methods in constrained nonlinear optimization, by drawing connections to insights from past work on perfect-recall two-player zero-sum games.
5. We added Proposition 4 to Section 3.2 that shows that **regret matching methods are guaranteed to converge to the optimal strategy (resp. global maximum) in best iterate if the utility function (resp. polynomial objective function) is concave**. This theoretical guarantee is known to hold for any regret minimizer, and yet, it can serve as a theoretical motivation—as asked for by Reviewer YU3n—to test whether “RM can act as a ‘first-order optimizer.’ ” Nonetheless—and as we already emphasized in our original submission—we agree that proving that RM methods are guaranteed to converge to a first-order optimal point in general nonconcave settings is an exciting avenue for future research, arguably, ignited by the practical insights from our paper. As it turns out, even a special case of this problem—convergence of RM to Nash equilibria in identical-interest games—has been open for more than 20 years (at least, since Hart and Mas-Colell, 2003), so resolving that is challenging. Absentmindedness—which is the focus of our paper—makes the problem even harder by making variables in the utility function appear in higher polynomial degree.

---

> ### Author Response · Authors · 2025-12-03
> **Part 2, and a concise summary of our contributions**
>
> Continuing the points above:
>
> 6. We improved our description of the subgroup detection problems under privacy constraints in Section 4.2 (per Reviewer rBEf’s feedback).
> 7. As per Reviewer pAUu’s suggestion, we added the implementation details of our polynomial optimization program in Gurobi in Appendix Section A.3.
> 8. We added **Table 2** which reports **aggregate statistics of our experimental results** on the full benchmark suite (due to Reviewer pAUu’s suggestion towards more readability), and updated the shaded regions in Figure 4 to cover the full min-to-max range of our runs (suggested by rBEf). Neither affect our experimental insights.
>
>
> ### Last but not least,
> We would like to summarize again the **two novel and valuable contributions** that this work provides.
> 1. **Benchmark Suite:** Decision making under imperfect recall, and specifically absentmindedness, have been extensively studied since the early years of game theory, and so far, this has been done with pen and paper. Our work is the first to develop an empirical framework for decision making under (arbitrary forms of) imperfect recall through a flexible suite of benchmarks and implemented algorithms. Earlier empirical work from the literature on game abstractions focused only on narrow subclasses of imperfect recall, while our scope here is much broader.
> 2. **Algorithmic Contribution:** Our work is the first to apply regret matching methods to general imperfect-recall problems, and more broadly, nonlinear optimization problems in practice. This is in contrast to perfect-recall (two-player) zero-sum games in which RM has been widely popular. Our main algorithmic contribution lies in extending the scope of RM methods to any imperfect-recall problem (by applying RM to the _agent-form_ of the decision problem), and showing that RM and RM+ attain strong performance in nonlinear optimization. The fact that they perform so well in our benchmarks is surprising, since we are working with a class of problems that is fundamentally harder than perfect-recall two-player zero-sum games (cf. Theorem 1 and Proposition 2).

---

### Meta-Review · Area_Chair_Ay1t · 2026-01-05

**Summary:**

Strength: The paper introduces a benchmark suite and through extensive empirical study, it shows that the family of RM algorithms can outperform first-order methods.

Weakness: The theoretical contribution appears limited, and most findings are supported only empirically. The technical novelty also needs to be better highlighted.

**Reviewer Concerns:**

Weakness: The theoretical contribution appears limited, and most findings are supported only empirically. The technical novelty also needs to be better highlighted.

**Reviewer Scores:**

Based on the comments and discussions, the reviewers will likely maintain their scores.

---

### Decision · Program_Chairs · 2026-01-26

Reject